# Progressive axonopathy when oligodendrocytes lack the myelin protein CMTM5

Tobias J Buscham[1], Maria A Eichel-Vogel[1], Anna M Steyer[1,2], Olaf Jahn[3,4], Nicola Strenzke[5], Rakshit Dardawal[6], Tor R Memhave[6], Sophie B Siems[1], Christina Müller[7], Martin Meschkat[1,8], Ting Sun[1], Torben Ruhwedel[1,2], Wiebke Möbius[1,2], Eva-Maria Krämer-Albers[7], Susann Boretius[6], Klaus-Armin Nave[1], Hauke B Werner[1]*

[1]Department of Neurogenetics, Max Planck Institute of Experimental Medicine, Göttingen, Germany; [2]Electron Microscopy Core Unit, Max Planck Institute of Experimental Medicine, Göttingen, Germany; [3]Proteomics Group, Max Planck Institute of Experimental Medicine, Göttingen, Germany; [4]Translational Neuroproteomics Group, Department of Psychiatry and Psychotherapy, University Medical Center Göttingen, Georg-August-University, Göttingen, Germany; [5]Institute for Auditory Neuroscience, University Medicine Göttingen, Göttingen, Germany; [6]Functional Imaging Laboratory, German Primate Center, Leibniz Institute for Primate Research, Göttingen, Germany; [7]Institute of Developmental Biology and Neurobiology, Johannes Gutenberg University, Mainz, Germany; [8]Abberior Instruments Gmbh, Göttingen, Germany

*For correspondence:
hauke@em.mpg.de

**Abstract** Oligodendrocytes facilitate rapid impulse propagation along the axons they myelinate and support their long-term integrity. However, the functional relevance of many myelin proteins has remained unknown. Here, we find that expression of the tetraspan-transmembrane protein CMTM5 (chemokine-like factor-like MARVEL-transmembrane domain containing protein 5) is highly enriched in oligodendrocytes and central nervous system (CNS) myelin. Genetic disruption of the *Cmtm5* gene in oligodendrocytes of mice does not impair the development or ultrastructure of CNS myelin. However, oligodendroglial *Cmtm5* deficiency causes an early-onset progressive axonopathy, which we also observe in global and tamoxifen-induced oligodendroglial *Cmtm5* mutants. Presence of the *Wld*[S] mutation ameliorates the axonopathy, implying a Wallerian degeneration-like pathomechanism. These results indicate that CMTM5 is involved in the function of oligodendrocytes to maintain axonal integrity rather than myelin biogenesis.

## Editor's evaluation

This manuscript by Buscham and colleagues investigates the phenotype of mice lacking the myelin protein chemokine-like factor-like MARVEL-transmembrane domain containing protein 5 (CMTM5). Loss of CMTM5 has no detectable effect on myelin structure, but nevertheless leads to axonal degeneration. The data reveal previously undescribed functions for a myelin protein in preserving CNS axonal integrity. This article will be of particular interest to those in the field of myelin biology as well as to neuroscientists interested in mechanisms underlying axonal function and degeneration.

## Introduction

Myelination of axons by oligodendrocytes enables rapid, saltatory conduction of signals in the vertebrate central nervous system (CNS) (*Cohen et al., 2020*; *Hartline and Colman, 2007*; *Tasaki, 1939*). Additionally, oligodendrocytes support the long-term preservation of axons metabolically (*Fünfschilling et al., 2012*; *Lee et al., 2012*; *Philips et al., 2021*; *Saab et al., 2016*) and via extracellular vesicles (*Chamberlain et al., 2021*; *Frühbeis et al., 2020*; *Mukherjee et al., 2020*). In fact, myelin pathology in CNS disorders such as leukodystrophies, multiple sclerosis, and respective animal models is commonly associated with axonal degeneration (*Franklin et al., 2012*; *Stadelmann et al., 2019*; *Wolf et al., 2021*). Oligodendrocytes are thus required to maintain axonal integrity and ultimately CNS function. However, oligodendrocytes express thousands of transcripts (*Jäkel et al., 2019*; *Zhang et al., 2014*; *Zhou et al., 2020*) and myelin comprises hundreds of proteins (*Ishii et al., 2009*; *Jahn et al., 2020*), and our knowledge remains limited with respect to which molecules contribute to myelin biogenesis, axonal support, or both.

We recently found that a member of the chemokine-like factor-like MARVEL-transmembrane containing (CMTM) protein family, CMTM6, is expressed in Schwann cells, the myelinating cells of the peripheral nervous system (PNS), and that its deletion in mice affects the diameters and function of peripheral axons (*Eichel et al., 2020*). Based on this finding, we asked if a member of the CMTM family is expressed in oligodendrocytes, which may thus fulfill a similar function in the CNS. The CMTM protein family comprises eight members in humans (*Han et al., 2003*) that have mostly been associated with mediating tumor immunity (*Burr et al., 2017*; *Mezzadra et al., 2017*; *Shao et al., 2007*; *Xiao et al., 2015*; *Yuan et al., 2020*).

In this study, we focused on *Cmtm5* considering (i) its expression in oligodendrocytes according to bulk RNAseq data (*Zhang et al., 2014*), (ii) the finding that the *Cmtm5* gene promoter drives expression of Cre recombinase in cells designated as oligodendrocytes (*Gong et al., 2007*), and (iii) the mass spectrometric identification of chemokine-like factor-like MARVEL-transmembrane domain containing protein 5 (CMTM5) in CNS myelin (*Jahn et al., 2020*). By structure prediction, CMTM5 comprises four transmembrane domains with small intracellular N- and C-terminal domains and two small extracellular loops (*Jumper et al., 2021*) but, its name notwithstanding, no apparent chemokine-like sequence motif. Here, we assess the functional relevance of CMTM5 in oligodendrocytes. We find that CMTM5 is not required for normal myelination or axonal diameters in the CNS. However, our data indicate that CMTM5 is involved in the function of oligodendrocytes to maintain the integrity of CNS axons.

## Results

### Expression of CMTM5 is enriched in oligodendrocytes and CNS myelin

We explored the hypothesis that CNS myelin comprises a homolog of the recently identified (*Eichel et al., 2020*) PNS myelin protein CMTM6. Indeed, previous mass spectrometric analysis identified CMTM5 in myelin purified from the brains of C57Bl/6N mice (*Jahn et al., 2020*). In contrast, neither CMTM6 nor any other member of the protein family was detected in CNS myelin. Correspondingly, published RNA sequencing data *Zhang et al., 2014* demonstrate that mature oligodendrocytes (MOL) display substantial abundance of *Cmtm5* mRNA but not of any other gene family member (*Figure 1—figure supplement 1A*). Indeed, *Cmtm5* mRNA is enriched in oligodendrocytes, in which its abundance increases with differentiation from the progenitor (OPC) stage to the myelinating oligodendrocytes (MOL) stage (*Figure 1—figure supplement 1B*). When evaluating mRNA abundance according to published scRNA-seq data (*Jäkel et al., 2019*; *Zhou et al., 2020*), MOL (as annotated by high-level expression of myelin basic protein mRNA, *MBP/Mbp*) express *CMTM5/Cmtm5* mRNA in both humans and mice (*Figure 1—figure supplement 1C–F*).Additional to its expression in oligodendrocytes, astrocytes also display a low level of *CMTM5/Cmtm5* mRNA (*Figure 1—figure supplement 1B, C, E*).

To independently confirm CMTM5 as a myelin protein, we first used immunoblotting to assess its abundance in myelin biochemically purified from mouse brains in comparison to equal amounts of mouse brain lysate. One band was detected at the expected molecular weight of 18 kDa. Indeed, the abundance of CMTM5 was markedly higher in the myelin-enriched fraction than in brain lysates (*Figure 1A*), similar to the myelin markers PLP, CNP, and SIRT2. Confocal imaging of immunolabeled spinal cord sections revealed CMTM5 in CNS myelin of c57Bl/6N mice (*Figure 1B*). Importantly, no

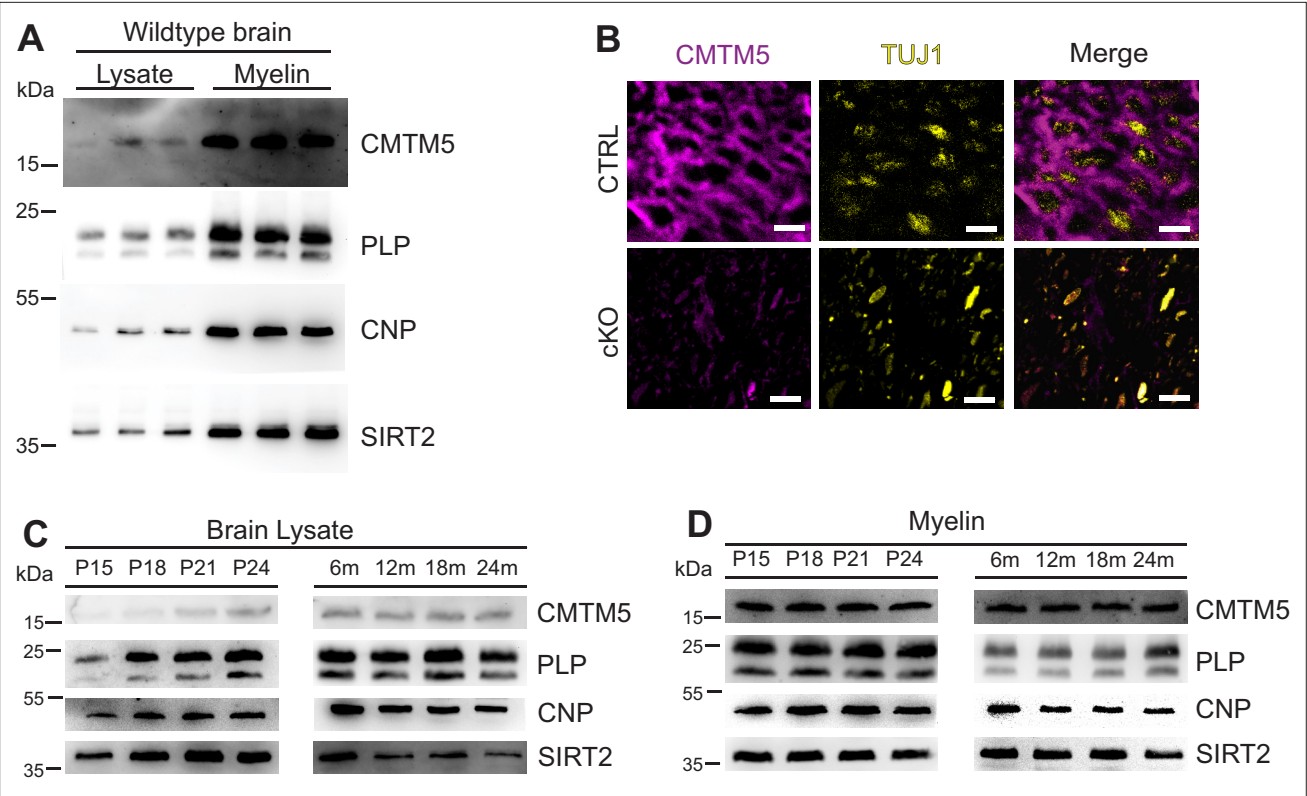

**Figure 1.** Identification of CMTM5 as a central nervous system (CNS) myelin protein. (**A**) Immunoblot analysis of CMTM5 in myelin biochemically purified from the brains C57/Bl6 mice at the age of 75 days (P75) compared to brain lysate with the same amount of protein loaded onto the gel. Note that CMTM5 is enriched in myelin. Known myelin proteins PLP, CNP, and SIRT2 are detected as markers. Shown are three biological replicates. (**B**) Immunohistochemistry and confocal microscopy of spinal cord sections of mice at P75. Note that CMTM5 (magenta) labeling was consistent with localization in myelin surrounding beta-III tubulin (TUJ1)-immunopositive axons (yellow) in CTRL (*Cmtm5^{fl/fl}*) mice. CMTM5 labeling was not detected in myelin of mice lacking *Cmtm5* expression in mature oligodendrocytes (*Cmtm5^{fl/fl};Cnp^{Cre/Wt}*, cKO). Scale bar, 2 μm. (**C, D**) Immunoblot analysis of CMTM5 in brain lysate (**C**) and biochemically purified myelin (**D**) of young and aged mice. Note that CMTM5 abundance in brain lysate increases coinciding with developmental myelination (**D**). Shown is one biological replicate per age. PLP, CNP, and SIRT2 were detected as markers. P, postnatal day; m, months.

The online version of this article includes the following source data and figure supplement(s) for figure 1:

Figure supplement 1. *Cmtm5* mRNA is expressed in mature oligodendrocytes (MOL) of mice and humans.

Figure supplement 1—source data 1. Numerical data for *Figure 1—figure supplement 1*.

labeling was found when analyzing corresponding sections of newly generated conditional mouse mutants with a deletion of the *Cmtm5* gene in myelinating cells (*Cmtm5^{fl/fl};Cnp^{Cre/Wt}*, also termed cKO; see below) (*Figure 1B*). By immunoblotting of homogenized wild-type mouse brains, the abundance of CMTM5 increased coinciding with myelin formation and maturation between postnatal days 15 (P15) and 24 (P24) (*Figure 1C*). In adult mouse brains between 6 and 24 months of age, the abundance of CMTM5 remained unchanged (*Figure 1C*). The abundance of CMTM5 in purified myelin also remained essentially constant (*Figure 1D*). Taken together, expression of CMTM5 in the CNS is highly enriched in mature oligodendrocytes and CNS myelin.

## CMTM5 is not essential for myelin biogenesis and composition

To assess the functional relevance of the expression of *Cmtm5* by oligodendrocytes, we generated mouse mutants with a conditional deletion of the gene selectively in myelinating cells (*Cmtm5^{fl/fl};Cnp^{Cre/Wt}*, also termed cKO). Conditional mutants were born at expected frequencies, and *Cmtm5* cKO mice showed no obvious behavioral phenotype. We biochemically purified myelin from brains of *Cmtm5* cKO mice and respective controls at P75 and further examined if CMTM5 deficiency affects the protein composition of myelin. By immunoblotting, CMTM5 was readily detected in myelin purified from the brains of control mice but undetectable in *Cmtm5* cKO myelin (*Figure 2A*). By label-free

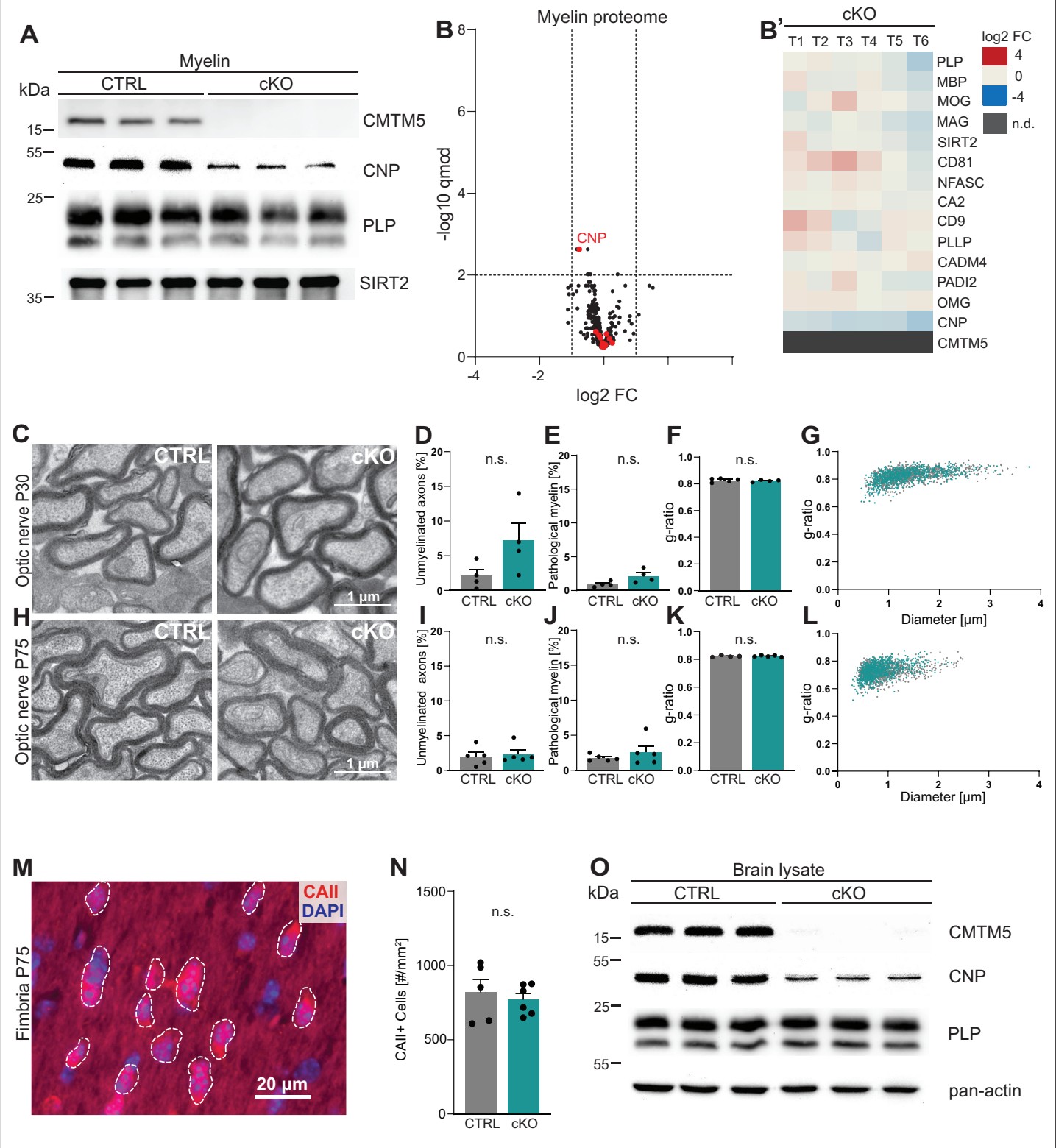

**Figure 2.** CMTM5 is not essential for myelin biogenesis and composition. (**A**) Immunoblot analysis shows that CMTM5 is undetectable in myelin purified from the brains of *Cmtm5*[fl/fl]*;Cnp*[Cre/Wt] (cKO) mice at postnatal day 75 (P75). CNP, PLP, and SIRT2 were detected as markers. Shown are three biological replicates per genotype. (**B, B'**) Quantitative proteome analysis of brain myelin reveals largely similar myelin composition in *Cmtm5* cKO and CTRL mice. Analyzed were n = 3 mice per genotype and two technical replicates per mouse (see *Figure 2—source data 1*). (**B**) Volcano plot with data points representing log2-fold change and -log10-transformed q-values of 428 identified proteins in *Cmtm5* cKO compared to CTRL myelin. Red

*Figure 2 continued on next page*

*Figure 2 continued*

dots highlight known myelin proteins. Stippled lines indicate thresholds. CMTM5 is not displayed because it was not identified in *Cmtm5* cKO myelin. (**B'**) Heatmap showing the relative abundance of selected known myelin proteins in *Cmtm5* cKO compared to control myelin. Data represents n = 3 mice per genotype analyzed as two technical replicates per mouse (T1–T6). Note that the relative abundance of most myelin proteins was essentially similar in *Cmtm5* cKO and CTRL myelin. In agreement with the immunoblot analysis in (**A**), the abundance of CNP was about halved in *Cmtm5* cKO myelin reflecting that the Cre driver line (*Cnp*$^{Cre/Wt}$) possesses only one *Cnp* allele. CMTM5 was not detected (n.d.) in *Cmtm5* cKO myelin. (**C–L**) Electron micrographs and quantitative assessment of myelin in CTRL and *Cmtm5*cKO optic nerves at postnatal day 30 (P30) (**C–G**) and P75 (**H–L**). Scale bar, 1 µm. Percentage of unmyelinated axons (**D, I**) and pathological myelin profiles is similar between the groups (**E, J**). Data correspond to all axons (on average more than 1500 axons) from 18 to 20 nonoverlapping random EM images from 4 to 5 animals per group. Two-tailed Student's *t*-test. (**D**) p=0.1003; (**E**) p=0.0598; (**I**) p=0.3937; (**J**) p=0.7269. Mean g-ratio (**F, K**) is similar between the experimental groups at P30 and P75. Data corresponds to 180–200 axons randomly selected from 18 to 20 EM images for each mouse. n = 4–5 mice per group. Two-tailed Student's *t*-test. (**F**) p=0.5839; (**K**) p=0.8821. (**G, L**) Scatter plot showing g-ratios in relation to respective axonal diameters. No apparent shift between the experimental groups is detectable. (**M, N**) Immunohistochemistry and genotype-dependent quantification of carbonic anhydrase 2 (CAII) immune-positive oligodendrocytes in a representative white matter tract (hippocampal fimbria) at P75. (**M**) Representative fluorescence micrograph, stippled lines encircle CAII-positive cells. Scale bar, 20 µm. (**N**) Number of CAII immunopositive cells number is similar in the fimbria of CTRL and *Cmtm5* cKO mice. n = 5–6 mice per group, unpaired Student's *t*-test p=0.5971. Bar graphs give mean ± SEM; data points in bar graphs represent individual mice. (**O**) Immunoblot analysis shows that CMTM5 is virtually undetectable in brain lysate of *Cmtm5*$^{fl/fl}$;*Cnp*$^{Cre/Wt}$ (cKO) mice at 12 months. CNP, PLP, and actin were detected as markers. Shown are three biological replicates per genotype.

The online version of this article includes the following source data and figure supplement(s) for figure 2:

**Source data 1.** Label-free quantification of proteins in CNS myelin fractions from Cmtm5 cKO and control mice.

**Source data 2.** Numerical data for *Figure 2*.

**Figure supplement 1.** Magnetic resonance imaging (MRI) of the brains of *Cmtm5* cKO mice.

**Figure supplement 1—source data 1.** Numerical data for *Figure 2—figure supplement 1*.

quantitative proteome analysis, *Cmtm5* cKO mice displayed a largely similar myelin proteome composition as control mice (*Figure 2B and B'*). As an exception, CMTM5 was undetectable in myelin purified from the brains of *Cmtm5* cKO mice, and the relative abundance of CNP was approximately halved as previously shown for the utilized *Cnp*$^{Cre/Wt}$ driver mice owing to heterozygosity of the *Cnp* gene (*Erwig et al., 2019*; *Lappe-Siefke et al., 2003*).

To examine if loss of CMTM5 affects the biogenesis and ultrastructure of myelin, we used transmission electron microscopy (TEM) to assess optic nerves dissected from *Cmtm5* cKO and control mice at P30 and P75 (*Figure 2C–L*). By morphometry, we did not observe signs of hypomyelination (*Figure 2D and I*) or myelin pathology such as myelin outfoldings, inner-tongue swellings, or lamella splittings (*Figure 2E and J*). The thickness of myelin sheaths, as determined by the g-ratio, was also virtually the same in *Cmtm5* cKO and control mice (*Figure 2F, G, K, L*). By immunohistochemistry, *Cmtm5* cKO mice displayed an unaltered number of cells immunopositive for carbonic anhydrase 2 (CAII), a marker for mature oligodendrocytes (*Figure 2M and N*). We then used magnetic resonance imaging (MRI) to assess the brains of 8-month-old *Cmtm5* cKO and respective control mice. However, no apparent differences in brain morphometry and diffusivity were found in various white and gray matter areas (*Figure 2—figure supplement 1*). Together, these data imply that expression of CMTM5 by oligodendrocytes is not essential for the normal biogenesis, ultrastructure, or protein composition of CNS myelin.

To determine to which extent cell types other than oligodendrocytes contribute to the expression of CMTM5 proteins in mature brains, we performed immunoblotting. CMTM5 was readily detected in whole-brain lysates of control mice but virtually undetectable in *Cmtm5* cKO whole-brain lysates (*Figure 2O*). This implies that non-oligodendroglial cells do not contribute substantially to expression of CMTM5 in mature brains.

## *Cmtm5* deletion in oligodendrocytes causes early-onset progressive axonopathy and late-onset general neuropathology

Considering that the diameters of peripheral axons are increased when Schwann cells lack CMTM6 (*Eichel et al., 2020*), we asked whether the diameters of CNS axons are altered when oligodendrocytes lack CMTM5. Yet, the quantitative assessment of TEMs did not reveal abnormal axonal diameters in the optic nerves of *Cmtm5* cKO mice (*Figure 3—figure supplement 1A and B*). Axonal

diameters were also normal in mice lacking CMTM5 in all cells (*Cmtm5*$^{-/-}$) compared to respective controls (*Figure 3—figure supplement 1C*).

However, in the course of this analysis we noted a considerable number of pathological-appearing axonal profiles. When quantifying these profiles at three different ages, we found their frequency to increase over time in the optic nerves of *Cmtm5* cKO compared to control mice (*Figure 3A and B*). Pathological profiles were evident in *Cmtm5* cKO optic nerves already at P30, and their number progressively increased toward 12 months of age, the oldest analyzed timepoint. We observed a trend toward a reduced density of axons in *Cmtm5* cKO mice at P30 and P75 that reached significance at 12 months of age (*Figure 3C*). This implies that the observed axonal pathology ultimately leads to axonal loss. To test whether *Cmtm5* cKO mice display pathological profiles also in other white matter tracts, we used electron microscopy to assess the dorsal white matter in spinal cords at 12 months of age. Indeed, the number of pathological profiles was increased several fold in *Cmtm5* cKO compared to control mice (*Figure 3D and E*), indicating that the observed axonopathy is not restricted to the optic nerve. A trend toward reduced axonal density in the spinal cord dorsal white matter of *Cmtm5* cKO at 12 months of age did not reach significance (*Figure 3F*). We presume that reduced axonal density would also gain significance in white matter tracts other than the optic nerve with further progression upon further aging.

We then used magnetic resonance spectroscopy (MRS) to determine the concentrations of the metabolites myo-inositol and N-acetyl-aspartate (NAA), which are considered neuropathological markers reflecting gliosis and axonal degeneration, respectively. We found the concentrations of myo-inositol significantly increased in the corpus callosi of *Cmtm5* cKO compared to control mice at 8 months of age (*Figure 3—figure supplement 2A*). *Cmtm5* cKO brains displayed a trend toward reduced concentrations of NAA, which did not reach significance (*Figure 3—figure supplement 2B*). We considered that these findings may imply the emergence of general neuropathology in the CNS of *Cmtm5* cKO mice, which we then aimed to resolve temporally. To this aim, we subjected the brains of *Cmtm5* cKO mice to immunohistochemistry at the ages of P30, P75, and 12 months. We immuno-labeled axonal swellings (using antibodies against APP), astrocytes (using antibodies against GFAP), and microglia (using the markers IBA1/AIF1 and MAC3/LAMP2). For quantification, we selected the hippocampal fimbria as a relatively uniform white matter tract. At age P30 and P75, we found no genotype-dependent differences between *Cmtm5* cKO and control mice with respect to number of APP-immunopositive axonal swellings or the relative area of immunopositivity for GFAP, IBA1, or MAC3. At 12 months of age, however, all markers were significantly increased in *Cmtm5* cKO compared to control mice (*Figure 3—figure supplement 3I–L*).

## Pathology of axon/myelin units by focused ion beam-scanning electron microscopy

Considering that two-dimensional visualization allows only limited insight into morphological features, we next assessed pathological profiles in the optic nerves of 12-month-old *Cmtm5* cKO mice using three-dimensional reconstruction of datasets gained by focused ion beam-scanning electron micros-copy (FIB-SEM; *Figure 4*). We found that the numbers of myelin outfoldings, inner-tongue inclu-sions, and axoplasmic inclusions are not increased in *Cmtm5* cKO mice (*Figure 4A''–C''*). Interestingly, however, the number of myelin whorls is markedly increased in *Cmtm5* cKO mice (*Figure 4D''*). Myelin whorls are multilamellar structures that largely display the periodicity of CNS myelin devoid of a discernible axon, probably best interpreted as remnants of degenerating myelinated fibers with rela-tive sparing of myelin membranes (*Edgar et al., 2009*). These data indicate that axonal degeneration – but not myelin pathology such as myelin outfoldings or inner-tongue inclusions – emerges when oligodendrocytes lack CMTM5.

## Functional assessment of retinae and optic nerves

As a read-out for visual function, we first assessed retinal function by electroretinography (ERG) recordings from *Cmtm5* cKO and control mice at 8.5 months of age. ERG waveforms (*Figure 5A*), ERG thresholds (*Figure 5B*), and the amplitudes of the a- and b-waves (*Figure 5C and D*) did not differ between the genotypes, indicating normal retinal function. However, visually evoked potentials (VEPs) implied that transmission of signals via the optic nerves to the visual cortex was impaired in *Cmtm5* cKO mice (*Figure 5E–H*). All mice displayed sizeable VEPs (*Figure 5E*) with normal thresholds

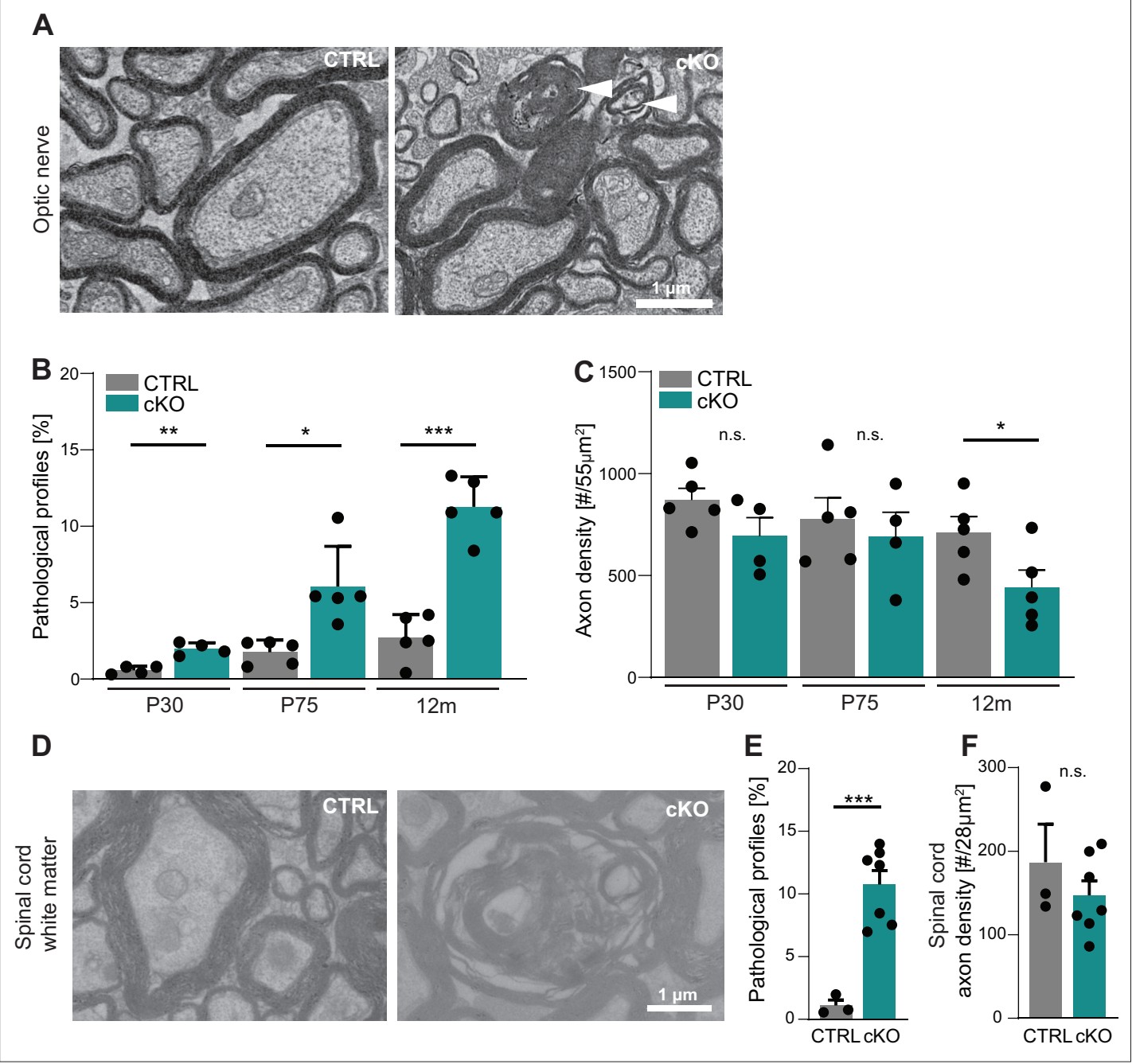

**Figure 3.** *Cmtm5* deletion in oligodendrocytes causes axonopathy. (**A, B**) Electron micrographs and genotype-dependent quantitative assessment of CTRL and *Cmtm5* cKO optic nerves at postnatal day 75 (P75). Scale bar, 1 µm. (**A**) Arrowheads point at pathological axons. (**B**) Quantification of pathological profiles reveals progressive axonopathy in optic nerves of *Cmtm5* cKO mice. n = 4–5 mice per group, 18–20 random nonoverlapping electron micrograph images analyzed, Two-tailed Student's *t*-test. Postnatal day 30 (P30) p=0.0011; P75 p=0.0191 with Welch's correction; 12 months p<0.0001. (**C**) Quantitative assessment of axonal density on semithin optic nerve sections. n = 4–5 mice per group, data represents mean axon number in five 55 µm² rectangles per mouse randomly distributed over the entire optic nerve. Axon numbers are significantly reduced at 12 months of age according to two-tailed Student's *t*-test. P30 p=0.1288; P75 p=0.5993; 12 months p=0.0499. (**D–F**) Electron micrographs and genotype-dependent quantitative assessment of spinal cord dorsal white matter in CTRL and *Cmtm5* cKO mice at 12 months. Scale bar = 1 µm. (**E**) Number of pathological profiles is increased in spinal cord dorsal white matter of *Cmtm5* cKO mice at 12 months. n = 3–7 mice per group, 16–20 14.84 µm² rectangles per mouse randomly distributed over the spinal cord dorsal white matter. (**F**) Axonal density assessed on electron micrographs of spinal cord dorsal white matter at 12 months. Trend toward reduced axon numbers in *Cmtm5* cKO did not reach significance according to two-tailed Student's *t*-test p=0.3341. n = 3–7 mice per group, data represents mean axon number in 5–728 µm² rectangles per mouse randomly distributed over the spinal cord dorsal white matter. All bars show mean ± SEM; all data points represent individual mice.

*Figure 3 continued on next page*

*Figure 3 continued*

The online version of this article includes the following source data and figure supplement(s) for figure 3:

**Source data 1.** Numerical data for *Figure 3*.

**Figure supplement 1.** *Cmtm5* deletion in oligodendrocytes does not affect the calibers of healthy-appearing axons.

**Figure supplement 1—source data 1.** Numerical data for *Figure 3—figure supplement 1*.

**Figure supplement 2.** Magnetic resonance spectroscopy (MRS) of the corpus callosum of *Cmtm5* cKO mice.

**Figure supplement 2—source data 1.** Numerical data for *Figure 3—figure supplement 2*.

**Figure supplement 3.** Secondary neuropathology following *Cmtm5* deletion.

**Figure supplement 3—source data 1.** Numerical data for *Figure 3—figure supplement 3*.

(*Figure 5F*) and a normal VEP latency (*Figure 5G*), indicating normal speed of action potential propagation and probably reflecting normal myelination in the optic nerves. However, the VEP amplitudes were significantly reduced in *Cmtm5* cKO mice (*Figure 5H*), probably owing to the axonopathy (*Figures 3 and 4*).

## Axonopathy in constitutive and tamoxifen-induced *Cmtm5* mutants

All results presented thus far are based on the analysis of mice in which the *Cmtm5* allele was recombined by Cre expressed in myelinating cells under the control of the *Cnp* promoter. Importantly, the utilized heterozygous $Cnp^{Cre/Wt}$ driver mice (*Lappe-Siefke et al., 2003*) harbor only one functional *Cnp* allele. Notwithstanding that heterozygous $Cnp^{Cre/Wt}$ mice display neuropathology only at old age (*Hagemeyer et al., 2012*), we sought to test if *Cmtm5* mutant mice also display axonopathy on a homozygous wild-type *Cnp* gene background. To this aim, we bred mice carrying a homozygous deletion of the *Cmtm5* gene in all cells ($Cmtm5^{-/-}$, knock-out; $Cmtm5^{Wt/Wt}$, control). As expected, CMTM5 was readily detectable by immunoblot in myelin purified from the brains of control mice but undetectable in $Cmtm5^{-/-}$ myelin. Importantly, the abundance of CNP appeared similar in $Cmtm5^{-/-}$ and $Cmtm5^{Wt/Wt}$ control myelin (*Figure 6A*), as was that of the myelin marker SIRT2. Vice versa, the abundance of CMTM5 appeared similar by immunoblot analysis of purified from the brains of $Plp^{-/Y}$ and $Cnp^{-/-}$ and respective control mice (*Figure 6—figure supplement 1A and B*). Thus, the abundance of PLP and CNP in myelin does not depend on CMTM5 and vice versa the abundance of CMTM5 in myelin does not depend on PLP or CNP.

We then used conventional TEM to scrutinize optic nerves dissected from $Cmtm5^{-/-}$ mice. Notably, by quantitative assessment of electron micrographs we found a progressive increase in the number of pathological profiles in $Cmtm5^{-/-}$ compared to control mice (*Figure 6B and C*), in similarity to *Cmtm5* cKO mice (*Figure 3*). Importantly, this indicates that the axonopathy that emerges when oligodendrocytes lack CMTM5 is independent of *Cnp* heterozygosity.

To rule out that the pathological profiles in *Cmtm5* mutant mice are the consequence of subtle developmental defects, we used the $Plp^{CreERT2}$ driver line (*Leone et al., 2003*) to induce recombination of the *Cmtm5* gene by injecting tamoxifen into adult $Cmtm5^{fl/fl};Plp^{CreERT2}$ mice (termed *Cmtm5* iKO in the following). Tamoxifen-injected $Cmtm5^{fl/fl}$ mice served as controls. By immunoblot, the abundance of CMTM5 was greatly reduced in myelin purified from the brains of *Cmtm5* iKO mice 4 months after tamoxifen injection (*Figure 6D*). Importantly, by quantitative assessment of electron micrographs, *Cmtm5* iKO mice displayed a significantly increased number of pathological profiles 4 months after tamoxifen injection (*Figure 6E and F*). This indicates that continued oligodendroglial expression of CMTM5 in adult mice is required to prevent the emergence of axonopathy.

## Axonopathy upon *Cmtm5* deletion is counteracted by the Wallerian degeneration slow (*Wld*S) mutation

To test if the axonopathy in *Cmtm5* mutants causes a decline in the number of neuronal cell bodies, we quantified retinal ganglion cells (RGCs) in the retinae of *Cmtm5* cKO and control mice at 12 months of age (*Figure 7A–C*). We found that RGC numbers were similar, indicating that neuronal cell bodies are preserved. Considering that this finding may imply a Wallerian-type pathomechanism of axon degeneration (*Coleman and Höke, 2020*), we assessed if the presence of the $Wld^S$ mutation (*Coleman et al., 1998*; *Lunn et al., 1989*) affects the number of pathological profiles upon *Cmtm5* deficiency. Indeed,

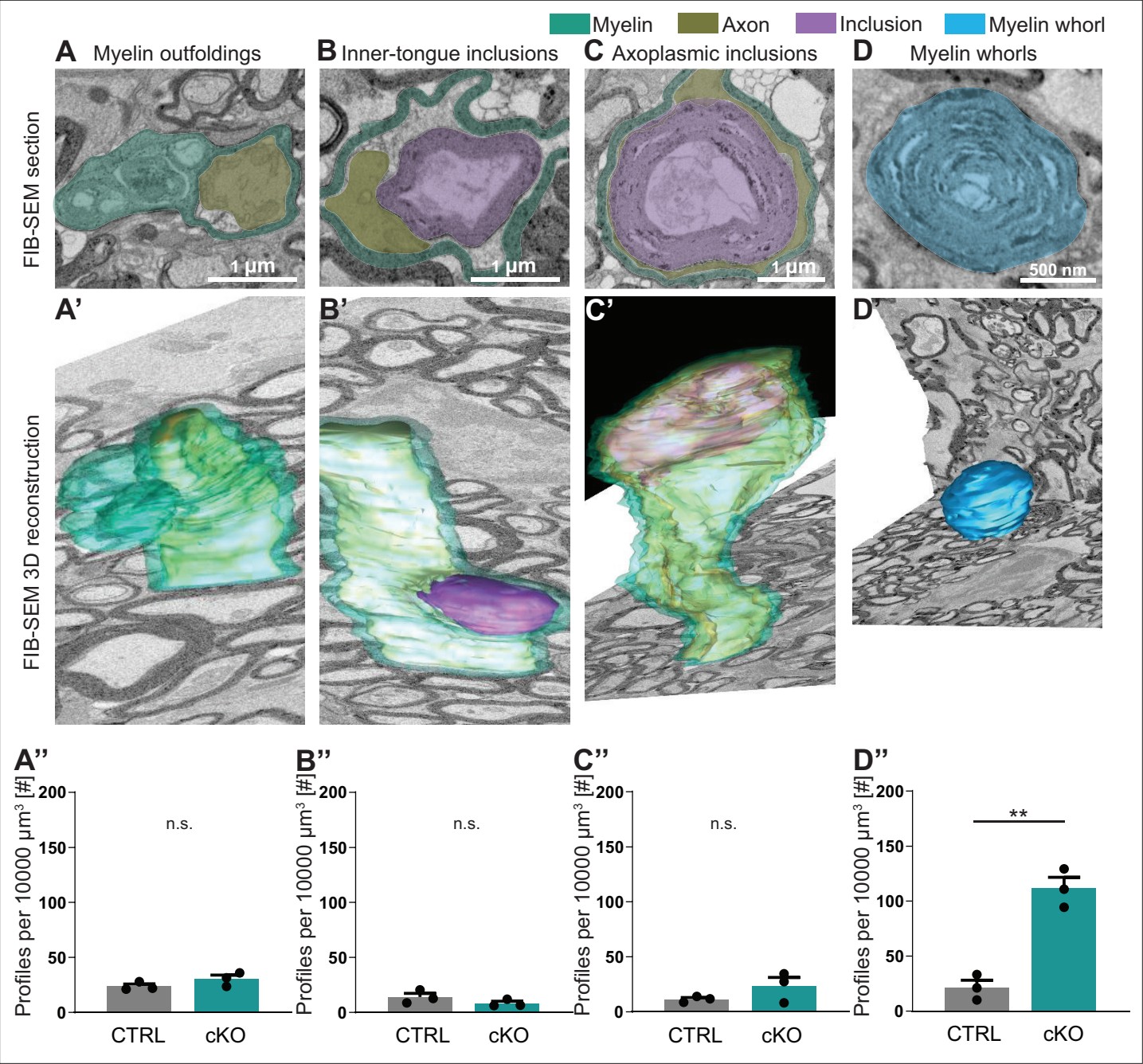

**Figure 4.** Focused ion beam-scanning electron microscopy (FIB-SEM) analysis specifies pathological profiles in *Cmtm5* cKO mice. FIB-SEM micrographs (**A–D**) and three-dimensional (3D) reconstruction (**A'–D'**) of pathological profiles in *Cmtm5* cKO optic nerve at 12 months of age. (**A–C**) Scale bar = 1 µm. (**D**) Scale bar = 500 nm. Myelin (cyan), axons (gold), inclusion (purple), and myelin whorls (blue) are highlighted. Pathological profiles include myelin outfoldings, inclusions in the inner tongue, inclusions completely engulfed by axoplasm, and myelin whorls. Analysis of the entire 3D volumes reveals that the relative number of myelin whorls is significantly increased in *Cmtm5* cKO mice. FIB-SEM stacks of optic nerves of three mice per genotype were analyzed. Normalized volume = 10,000 µm³. Two-tailed Student's *t*-test. (**A''**) p=0.1882; (**B''**) p=0.2190; (**C''**) p=0.2111; (**D''**) p=0.0017. Scale bars 1 µm in (**A–C**), 500 nm in (**D**). Bar graphs give mean ± SEM.

The online version of this article includes the following source data for figure 4:

**Source data 1.** Numerical data for *Figure 4*.

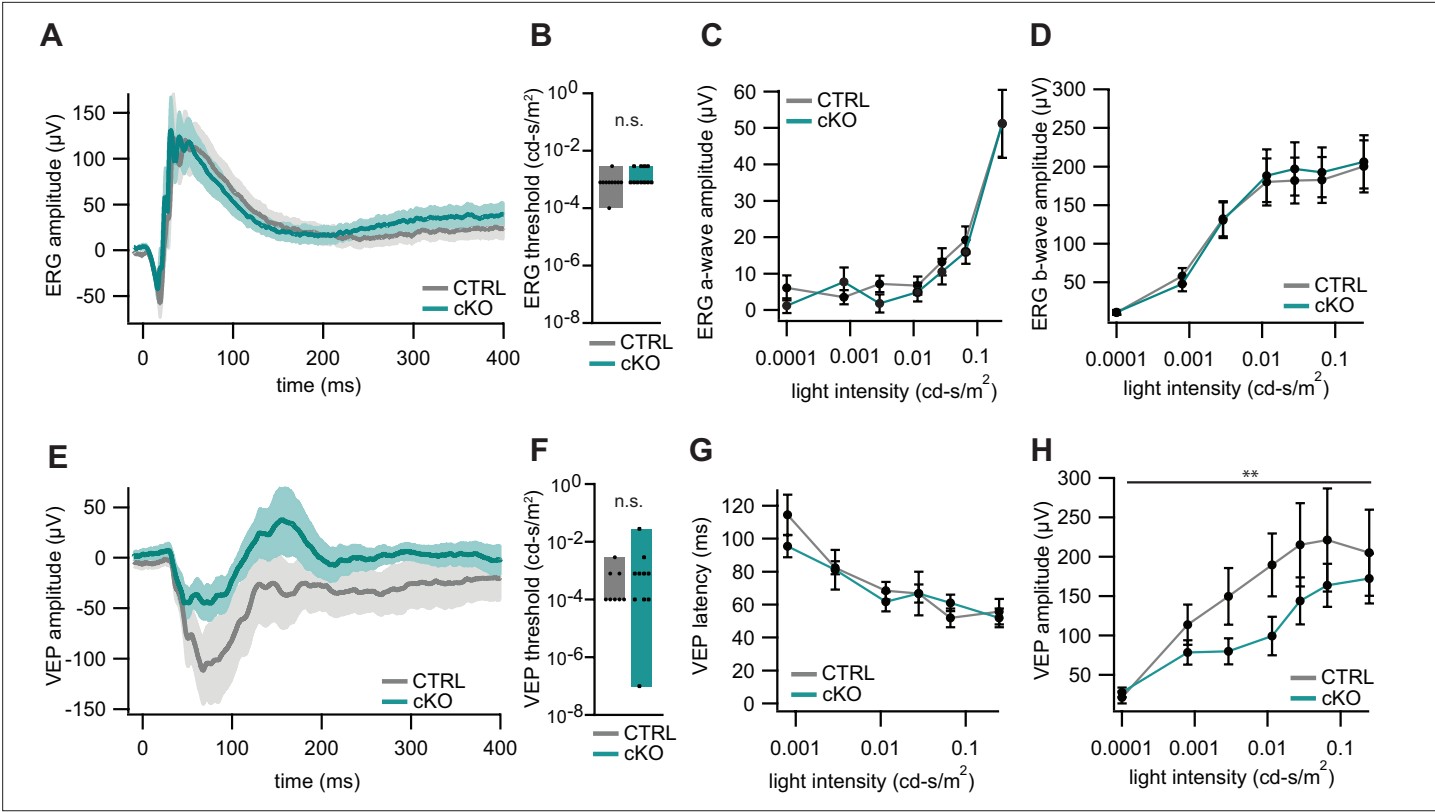

**Figure 5.** Electroretinography (ERG) and visually evoked potentials (VEPs) of *Cmtm5* cKO mice. (**A–D**) ERG. (**A**) ERG waveforms in response to light flashes at 0.25 cd-s/m² from 11 *Cmtm5* cKO (grand average turquoise, SEM shaded) and 10 CTRL mice (grand average gray, SEM shaded). (**B**) ERG thresholds are similar between CTRL and *Cmtm5* cKO. Unpaired Student's *t*-test of the mean ± SEM p=0.13. (**C, D**) Amplitudes of the ERG a- and b-waves in response to light flashes of varying intensities in *Cmtm5* cKO (n = 11, turquoise) and CTRL mice (n = 11, gray; mean ± SEM) are similar between genotypes. Two-way ANOVA. (**C**) p=0.42, (**D**) p=0.79. Analysis was performed at 8.5 months of age. (**E–H**) VEPs. (**E**) VEP in response to light flashes at 0.01 cd-s/m² from 10 *Cmtm5* cKO (grand average turquoise, SEM shaded) and 9 CTRL mice (grand average gray, SEM shaded) display comparable waveforms dominated by a broad negative wave in both genotypes. (**F**) VEP thresholds are not significantly different between CTRL and *Cmtm5* cKO. Unpaired Student's *t*-test of the mean ± SEM with Welch's correction p=0.33. (**G, H**) VEP latencies and amplitudes in response to light flashes of varying intensities in *Cmtm5* cKO (n = 10, turquoise) and CTRL mice (n = 8, gray; means ± SEM). Note that *Cmtm5* cKO and CTRL mice show similar VEP latencies but *Cmtm5* cKO mice display reduced VEP amplitudes compared to CTRL mice. Two-way ANOVA. (**G**) p=0.61, (**H**) p=0.005. Analysis was performed at 8.5 months of age.

The online version of this article includes the following source data for figure 5:

**Source code 1.** Code used for analysis of visually evoked potential (VEP) and electroretinography (ERG) in *Figure 5*.

**Source data 1.** Numerical data for *Figure 5*.

when we analyzed the optic nerves of *Cmtm5*⁻/⁻ mice by TEM at the age of 6 months, heterozygous presence of the *Wld^S* mutation markedly reduced the number of pathological profiles (*Figure 7D and E*). For comparison, *Cmtm5^Wt/Wt* mice displayed only a negligible number of pathological profiles, independent of the presence of the *Wld^S* mutation. Together, these results imply a Wallerian-type pathomechanism of axonopathy when oligodendrocytes lack *Cmtm5*.

## Discussion

We report the intriguing observation that mice lacking the CNS myelin protein CMTM5 display an early-onset progressive axonopathy, whereas the biogenesis and ultrastructure of myelin appear unaffected. According to previously established datasets, expression of *Cmtm5* in the CNS is highly enriched in myelinating oligodendrocytes (*Jäkel et al., 2019*; *Zhang et al., 2014*; *Zhou et al., 2020*). CMTM5 is not the first myelin protein associated with secondary axonal degeneration in mutant mice. However, different from the previously studied myelin genes *Plp1* (*Edgar et al., 2004*; *Griffiths et al.,*

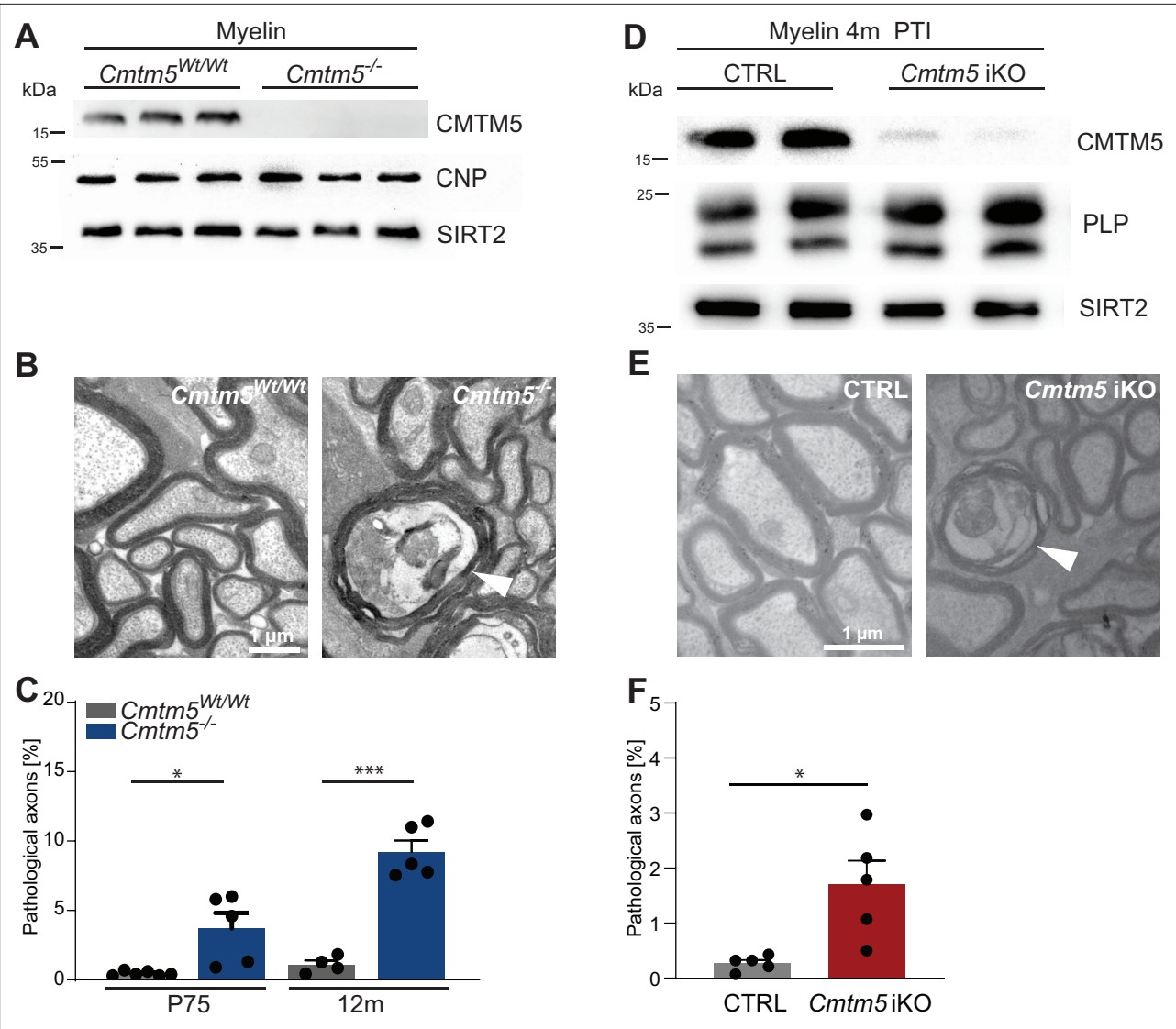

**Figure 6.** Axonopathy in constitutive and tamoxifen-induced *Cmtm5* mutants. (**A–C**) Analysis of mice lacking *Cmtm5* expression from all cells (*Cmtm5⁻/⁻* mice) and respective controls. (**A**) Immunoblot confirms absence of CMTM5 in myelin purified from the brains of *Cmtm5⁻/⁻* mice. CNP and SIRT2 were detected as controls. (**B**) Representative electron micrographs (EMs) of *Cmtm5⁻/⁻* and respective control optic nerves. Arrowhead points at pathological profile. Scale bar, 1 µm. (**C**) Quantitative assessment of 18–20 random nonoverlapping EMs from 4 to 6 mice per group. Note the progressive increase in pathological appearing axons in optic nerves of *Cmtm5⁻/⁻* mice. Postnatal day 75 (P75): p=0.0406 by two-sided Student's *t*-test with Welch's correction; postnatal day 365 (P365): p<0.0001 two-sided Student's *t*-test. (**D–F**) Analysis of mice lacking *Cmtm5* expression in mature oligodendrocytes upon induction by tamoxifen (*Cmtm5ᶠˡ/ᶠˡ;Plpᶜʳᵉᴱᴿᵀ²*, iKO) and respective tamoxifen-injected *Creᴱᴿᵀ²*-negative controls (*Cmtm5ᶠˡ/ᶠˡ*, CTRL). (**D**) Immunoblot of myelin purified from the brains of mice 4 months post tamoxifen injection (PTI). Note that the abundance of CMTM5 is strongly reduced in *Cmtm5* iKO myelin. PLP and SIRT2 were detected as controls. (**E**) Representative EMs of *Cmtm5* iKO and CTRL optic nerves. Arrowhead points at pathological profile. Scale bar, 1 µm. (**F**) Quantification of pathological profiles (20 nonoverlapping random images per mouse, n = 5 mice per genotype). Number of pathological profiles is significantly increased 4 months PTI (p=0.0282 by two-sided Student's *t*-test with Welch's correction). Bar graphs give mean ± SEM; data points represent individual mice.

The online version of this article includes the following source data and figure supplement(s) for figure 6:

**Source data 1.** Numerical data for *Figure 6*.

**Figure supplement 1.** Absence of evidence that the abundance of CMTM5 is altered in central nervous system (CNS) myelin of *Plp*- or *Cnp*-deficient mice.

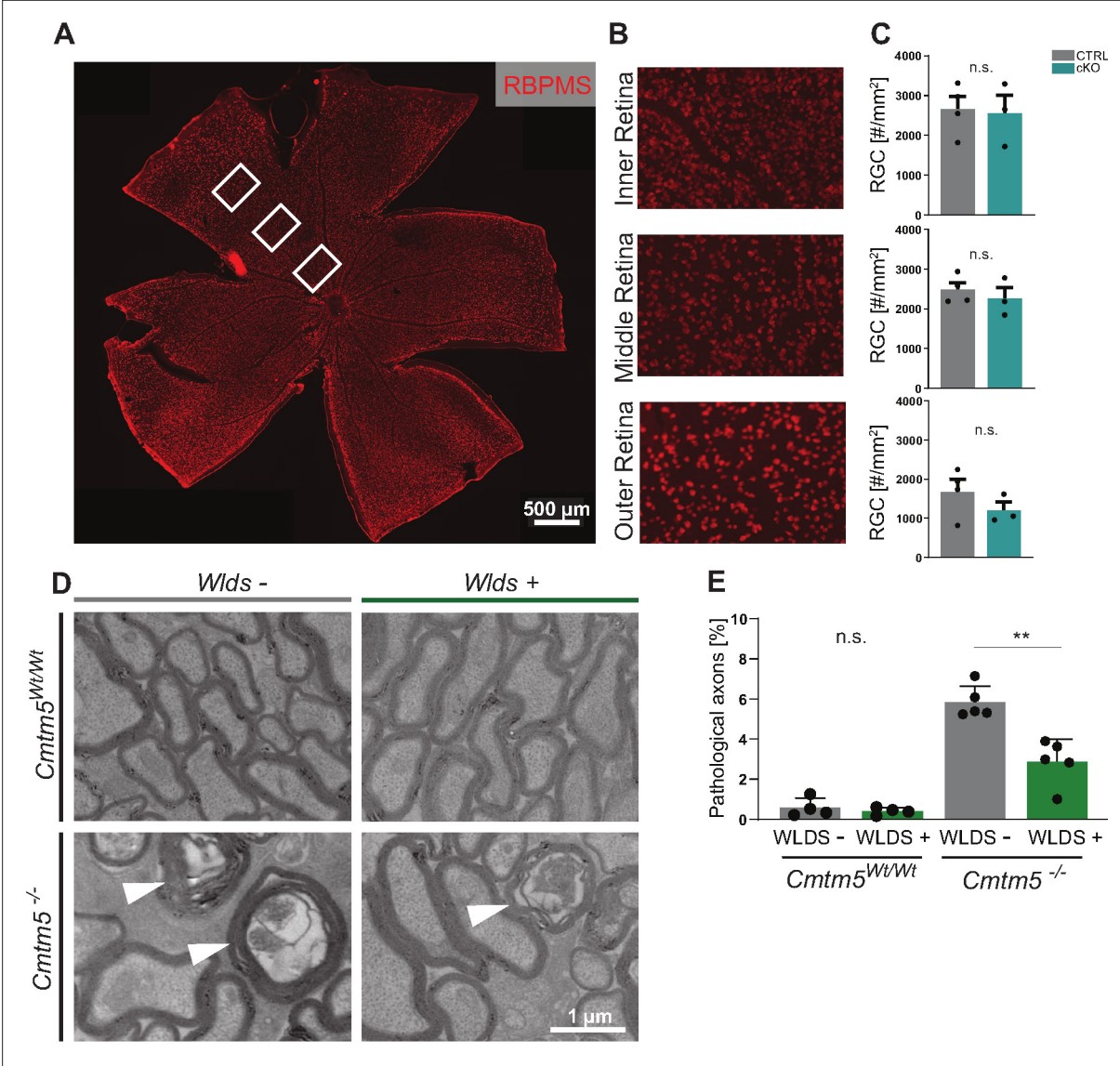

**Figure 7.** Axonopathy upon *Cmtm5* deletion counteracted by the *Wld$^S$* mutation. (**A**) Retinae dissected from *Cmtm5* cKO and CTRL mice were immunolabeled with antibodies detecting RBPMS as a marker for retinal ganglion cells (RGCs). Image representative of n = 3 retinae. Scale bar = 500 μm. (**B**) Magnification of the inner, middle, and outer parts of the retina. (**C**) Quantitative assessment indicates that the number of RGCs is similar between *Cmtm5* cKO and CTRL mice. Retinae of 12-month-old mice were analyzed. Data represent the mean of three nonoverlapping areas assessed for each zone (inner, middle, outer retina), as indicated by the white boxes in (**A**). Unpaired two-sided Student's *t*-test inner part: p=0.8484; middle part: p=0.5211; outer part: p=0.2912. (**D, E**) Electron micrographs (EMs) and genotype-dependent quantification of pathological profiles in the optic nerves of *Cmtm5$^{-/-}$* and *Cmtm5$^{Wt/Wt}$* control mice in dependence of the heterozygous presence of the *Wld$^S$* mutation (symbolized *Wld$^S$*+) at 6 months of age. Representative EMs; arrowheads point at pathological profiles. Scale bar, 1 μm. (**E**) Quantification of pathological profiles in the optic nerves of 6-month-old mice. Note that *Cmtm5* deletion causes an increased number of pathological profiles, which is reduced by the presence of the *Wld$^S$* mutation (symbolized *Wld$^S$*+). Data correspond to optic nerves from 4 to 5 mice per group and 20 random nonoverlapping EM images analyzed. Unpaired two-sided Student's *t*-test *Cmtm5$^{Wt/Wt}$*: p=0.5107; *Cmtm5$^{-/-}$*: p=0.0014. Bar graphs give mean ± SEM; data points represent individual mice.

The online version of this article includes the following source data for figure 7:

**Source data 1.** Numerical data for *Figure 7*.

*1998*; *Lüders et al., 2019*) and *Cnp* (*Edgar et al., 2009*; *Lappe-Siefke et al., 2003*), the encoded protein is of much lower abundance in the myelin sheath. By quantitative mass spectrometry, CMTM5 represents only 0.027% of the myelin proteome (*Jahn et al., 2020*), in comparison to 37.9% for PLP and 5.1% for CNP. The relative abundance of CMTM5 in myelin is thus roughly equivalent to that of other transmembrane-tetraspan proteins CD9 (0.06%), proteolipid GPM6B (0.04%), and the gap junction

**Table 1.** Comparison of neuropathological features in *Cmtm5-*, *Cnp-,* and *Plp*-mutant mice.
Neuropathological features in *Cmtm5* cKO,*Cnp*[-/-], and *Plp*[-/Y] mice and key references are given. APP,
amyloid precursor protein.

| Feature | *Cmtm5* mutants | *Cnp* mutants | *Plp* mutants |
|---|---|---|---|
| Myelinated axons (%) | Normal | Reduced | Reduced |
| Myelin thickness | Normal | Trend to thinner myelin | Normal-appearing |
| Myelin structure | Normal | Inner-tongue swellings, myelin outfoldings | Lamella splittings, myelin outfoldings |
| Axonopathy | Early onset, progressive | Early onset, progressive | Early onset, progressive |
| Modified by *Wld*[S] | Reduction of pathology | No effect | No effect |
| APP[+] axonal swellings | Late onset | Early onset | Early onset |
| Microgliosis | Late onset | Early onset | Early onset |
| Astrogliosis | Late onset | Early onset | Early onset |
| References | This study | *Edgar et al., 2009*; *Lappe-Siefke et al., 2003*; *Patzig et al., 2016*; *Rasband et al., 2005* | *Edgar et al., 2004*; *Griffiths et al., 1998*; *Klugmann et al., 1997*; *Patzig et al., 2016* |

protein GJC3/Cx29 (0.02%) (*Jahn et al., 2020*), which were previously identified as low-abundant myelin constituents (*Kagawa et al., 1997*; *Kleopa et al., 2004*; *Werner et al., 2013*). Considering the low abundance of CMTM5 in myelin, it may not be unexpected that CMTM5-deficient mice do not display primary ultrastructural defects that affect the myelin sheath when highly abundant structural myelin proteins as PLP or CNP are lacking (*Table 1*). The comparison of our different mouse mutants has revealed that CMTM5 is required by mature oligodendrocytes. However, details of its mechanistic role in continued axon–glia interactions remain obscure.

CMTM5 is a member of the CMTM protein family (*Han et al., 2003*) that has been associated with regulating tumor immunity (*Burr et al., 2017*; *Mezzadra et al., 2017*; *Shao et al., 2007*; *Xiao et al., 2015*; *Yuan et al., 2020*), including CMTM5 itself. Comparatively little is known about the functional relevance of CMTM proteins in the nervous system. However, we recently found the paralog CMTM6 to be expressed in myelinating Schwann cells, in which it is involved in the previously unknown function of Schwann cells to restrict the diameters of peripheral axons (*Eichel et al., 2020*). The consequences of deleting CMTM5 and CMTM6 in oligodendrocytes and Schwann cells, respectively, may appear roughly similar when considering that the myelin ultrastructure is not affected while axonal features are altered. Notably, however, deleting CMTM6 from Schwann cells causes increased diameters of peripheral axons but no signs of actual degeneration (*Eichel et al., 2020*), whereas deleting CMTM5 from oligodendrocytes causes CNS axonopathy without altering axonal calibers. Thus, CMTM5 and CMTM6 expressed by oligodendrocytes and Schwann cells have distinct functions in the CNS and PNS, respectively.

Schwann cells, in addition to expressing CMTM6 (*Eichel et al., 2020*), also display considerable expression of CMTM5. In peripheral nerves, *Cmtm5*-mRNA display a developmental abundance profile similar to that of other peripheral myelin-related genes and strong enrichment in Schwann cells (*Gerber et al., 2021*; *Patzig et al., 2011*; *Siems et al., 2020*). By immunoblotting and cryo-immuno electron microscopy, CMTM5-protein localizes to peripheral myelin (*Patzig et al., 2011*). It will be a relevant future task to test its functional relevance in peripheral nerves by assessing mutant mice lacking expression in Schwann cells of either only CMTM5 or of both CMTM5 and CMTM6.

The CNS axonopathy observed upon deleting *Cmtm5* strongly implies that CMTM5 is involved in the oligodendroglial function of preserving axonal integrity and that this function is not limited to an early developmental stage. Oligodendroglial support of axons involves several mechanisms, including supplying energy-rich substrates via monocarboxylate transporters (*Fünfschilling et al., 2012*; *Lee et al., 2012*; *Philips et al., 2021*; *Trevisiol et al., 2020*), allocating antioxidative proteins and other enzymes via extracellular vesicles (*Chamberlain et al., 2021*; *Frühbeis et al., 2020*; *Mukherjee et al., 2020*), and modulating axonal transport (*Edgar et al., 2004*; *Frühbeis et al., 2020*), these mechanisms

being possibly interrelated. It will be an important next step to identify the specific mechanism(s) of axonal support that are impaired when oligodendrocytes lack CMTM5.

It is helpful to compare the pathology of previously described myelin mutants with the axonal defects in CMTM5-deficient mice as assessed here (*Table 1*). Notably, the presence of early-onset, progressive axonopathy is a shared feature of the myelin mutant mice lacking PLP (*Edgar et al., 2004*; *Griffiths et al., 1998*), CNP (*Edgar et al., 2009*; *Lappe-Siefke et al., 2003*), or CMTM5, which are normally myelinated (*Cmtm5* mutants) or display only moderate hypomyelination (*Plp1* and *Cnp* mutants). In contrast, entirely dysmyelinated *Mbp*-deficient *shiverer* mice (*Roach et al., 1985*) do not display axonal degeneration (*Griffiths et al., 1998*; *Ou et al., 2009*; *Uschkureit et al., 2000*). This indicates that the lack of myelin per se is less detrimental for axons than axonal ensheathment with functionally impaired myelin. Moreover, APP-immunopositive axonal swellings, astrogliosis, and microgliosis are early features when PLP (*Griffiths et al., 1998*; *de Monasterio-Schrader et al., 2013*; *Edgar et al., 2004*; *Steyer et al., 2020*; *Trevisiol et al., 2020*) or CNP (*Edgar et al., 2009*; *Lappe-Siefke et al., 2003*; *Wieser et al., 2013*) are lacking, but emerges at much older age in CMTM5-deficient mice. This indicates that the pathomechanisms differ between *Plp1* and *Cnp* mutants and *Cmtm5* mutants. This is supported by our observation that the axonopathy is ameliorated by the presence of the *Wld^S* mutation in CMTM5-deficient mice, different from that in PLP- and CNP-deficient mice, at least at the examined timepoints and in the presence of one copy of the *Wld^S* gene (*Edgar et al., 2004*; *Edgar et al., 2009*). Together, this suggests different dynamics, and probably different mechanisms of axonopathy, when oligodendrocytes lack PLP, CNP, or CMTM5.

The *Wld^S* mutation can protect axons from various types of physical, toxic, or genetic insult (*Coleman et al., 1998*; *Coleman and Höke, 2020*; *Lunn et al., 1989*). In the *Wld^S* pathway, a key regulator of axonal degeneration is the enzyme sterile alpha and TIR motif containing protein 1 (SARM1) (*Gerdts et al., 2013*; *Osterloh et al., 2012*). Both the *Wld^S* mutation and deletion of the *Sarm1* gene can delay axonal degeneration (*Coleman et al., 1998*; *Hopkins et al., 2021*), involving the maintenance of high NAD$^+$ levels along the axon (*Di Stefano et al., 2014*; *Gilley and Coleman, 2010*; *Hopkins et al., 2021*; *Wang et al., 2005*). To the best of our knowledge, CMTM5-deficient mice represent the first model in which the axonopathy that emerges upon deletion of a CNS myelin protein is ameliorated by the *Wld^S* mutation. On the other hand, the degeneration of axons in the PNS of mice lacking myelin protein zero (MPZ/P0) is robustly delayed by the presence of the *Wld^S* mutation (*Samsam et al., 2003*). Thus, genetic defects of either oligodendrocytes or Schwann cells can cause an axonopathy with a Wallerian-like pathomechanism in the CNS and PNS, respectively. It will be important to test if the currently developed small molecule SARM1 inhibitors (*Bosanac et al., 2021*; *Hughes et al., 2021*; *Loring et al., 2020*) allow counteracting axonal degeneration secondary to an insult primarily affecting oligodendrocytes.

## Materials and methods
### Mouse models and mouse lines
Frozen sperm of mice carrying the 'knockout-first' allele of the *Cmtm5* gene (C57BL/6N-A^tm1Brd^*Cmtm5*^tm1a(KOMP)Wtsi^/Wtsi) was acquired from The Mouse Genetics Project (Wellcome Trust Sanger Institute, Hinxton, UK). *Cmtm5*^tm1a(KOMP)Wtsi^ mice were generated by the transgene facility of the Max Planck Institute of Experimental Medicine (Göttingen, Germany) by in vitro fertilization using standard procedures. The LacZ/neo cassette was deleted by crossbreeding these mice with *Gt(ROSA)26Sor*^tm1(FLP1)Dym^ mice expressing flippase (*Farley et al., 2000*) yielding mice heterozygous for the floxed *Cmtm5* allele (*Cmtm5*^tm1c(KOMP)Wtsi^ mice, also termed *Cmtm5*^fl/Wt^), which were bred to homozygosity. To delete *Cmtm5* in oligodendrocytes, *Cmtm5*^fl/fl^ mice were crossbred with mice expressing *Cre* under the *Cnp* promoter (*Cnp1*^tm(cre puro)Kan^ mice, also termed *Cnp*^Cre/Wt^; *Lappe-Siefke et al., 2003*) yielding *Cmtm5*^fl/fl^;*Cnp*^Cre/Wt^ mice (also termed *Cmtm5* cKO). In experiments assessing *Cmtm5* cKO mice, *Cmtm5*^fl/fl^ mice served as controls. Taking advantage of germline-recombination, we gained a mouse line with a body-wide deletion of *Cmtm5* (*Cmtm5*^tm1d(KOMP)Wtsi^ mice, also termed *Cmtm5*^-/-^ or knock-out). In experiments assessing *Cmtm5*^-/-^ mice, *Cmtm5*^Wt/Wt^ mice served as controls. To delete *Cmtm5* in oligodendrocytes of adult mice, *Cmtm5*^fl/fl^ were crossbred with mice expressing tamoxifen-inducible *Cre* under the *Plp* promoter (Tg(Plp1-cre/ERT2)1Ueli, *Plp*^CreERT2^, *Leone et al., 2003*), resulting in *Cmtm5*^fl/fl^;*Plp*^CreERT2^ mice (also termed iKO) and respective controls without *Cre*. For induction, male mutant mice (*Cmtm5*^fl/^

*fl*;*Plp^CreERT2*) and male control mice (*Cmtm5^fl/fl*) were injected with tamoxifen intraperitoneally at 8 weeks of age for 10 days with a 2-day break after the first five injection days (1 mg tamoxifen dissolved in 100 µl corn oil per mouse and day). Heterozygous *Cmtm5^Wt/-* mice were crossbred with *Cmtm5^Wt/-* mice harboring the *Wld^S* mutation (*Coleman et al., 1998*; *Mack et al., 2001*) to obtain all experimental groups from the same breeding scheme (control groups: *Cmtm5^Wt/Wt* and *Cmtm5^Wt/Wt*;*Wld^S*; knock-out groups: *Cmtm5^-/-* and *Cmtm5^-/-*;*Wld^S*). Thus, the *Wld^S* allele was used heterozygously in all experiments in *Figure 7D and E*; *Wld^S*+ thus indicates heterozygosity for the *Wld^S* allele. *Wld^S*- indicates the absence of the *Wld^S* mutation. Littermate mice were used as experimental controls as far as possible.

*Cmtm5* mutant mice were generated in the C57BL/6N genetic background. *Rosa^FLP* mice, *Cnp^Cre* mice, and *Plp^CreERT2* mice were crossed for at least 10 generations into the C57BL/6N background before crossing with *Cmtm5* mutants. The experiments using *Cmtm5* mutant mice (*Figures 1–6*) can therefore be considered as in the C57BL/6N background. *Wld^S* mice were received designated as in the C57BL/6 background and crossed for one generation into the C57BL/6N background before crossing with *Cmtm5* mutant mice. It was not specifically tested if the experiments involving *Wld^S* mice (*Figure 7*) were in the C57BL/6N genetic background or in a hybrid background of C57BL/6N and C57BL/6J.

Genotyping was carried out by genomic PCR. *Cmtm5* genotypes were assessed with the same PCR strategy in all *Cmtm5* lines (*Cmtm5* cKO, *Cmtm5^-/-*,*Cmtm5* iKO, *Cmtm5*;*Wld^S*). Sense primer (5′-AGTAGTGGCCCATTGCCATC) in combination with antisense primer (5′-TGGTTAGGGG-GCTCCTCTTC) yielded a 626 bp product (floxed allele) or 437 bp product (wildtype). In the same reaction, antisense primer (5′-GAGCTCAGACCATAACTTCG) was used to detect *Cmtm5* allele recombination yielding a 313 bp fragment. Detection of the *Cnp^Cre* allele (*Lappe-Siefke et al., 2003*) was carried out using sense primer (5′-GCCTTCAAACTGTCCATCTC) and antisense primer (5′-CCCAG-CCCTTTTATTACCAC) amplifying a 700 bp product. As well as a sense (5′-CAGGGTGTTATAAGCA ATCCC) and antisense (5′- CCTGGAAAATGCTTCTGTCCG) primer yielding a 357 bp fragment when Cre positive. *Plp^CreERT2* (*Leone et al., 2003*) was detected using sense primer (5′-TGGACAGCTGGG ACAAAGTAAGC) and antisense primer (5′-CGTTGCATCGACCGGTAATGCAGGC) yielding a 250 bp product. The *Wld^S* mutation (*Coleman et al., 1998*; *Mack et al., 2001*) was detected using sense primer (5′-CGTTGGCTCTAAGGACAGCAC) and antisense primer (5′-CTGCAGCCCCCACCCCTT) yielding a 182 bp product.

Mice were bred and kept in the mouse facility of the Max Planck Institute of Experimental Medicine, Göttingen. Experimental mutant mice were analyzed with littermate controls as far as possible. All animal experiments were performed in accordance with the German animal protection law (Tier-SchG) and approved by the Niedersächsisches Landesamt für Verbraucherschutz und Lebensmittelsicherheit (LAVES) under license 33.19-42502-04-15/1833 and 33.8-42502-04-19/3172.

## Biochemical purification of myelin from mouse brains

Purification of a myelin-enriched light-weight membrane fraction from nervous tissue using sucrose density centrifugation and osmotic shocks was previously described (*Erwig et al., 2019*). Mice were sacrificed by cervical dislocation. Protein concentrations of brain lysate and myelin fractions were determined using the DC Protein Assay Kit (Bio-Rad, Munich, Germany) following the manufacturer's instruction and measured using the Eon High Performance Microplate Spectrophotometer (BioTek, Vermont, USA).

## Immunoblotting

Immunoblotting was essentially performed as described (*Schardt et al., 2009*). Brain lysate and myelin fraction samples were diluted in 4× sodium dodecyl sulfate (SDS) sample buffer (glycerol 40% [w/v], Tris/HCl pH 6.8 240 mM, SDS 8% [w/v] bromophenol blue 0.04% [w/v]); 5% dithiothreitol (DTT) was added as a reducing agent. Before usage, samples were heated at 40°C for 10 min. For protein separation by SDS-PAGE, the Mini-PROTEAN Handcast system (Bio-Rad, Munich, Germany) was used with self-casted acrylamide gels (10–15%). 5–15 µg samples were loaded per well (depending on protein of interest) next to 5 µl pre-stained protein ladder (PageRuler, Thermo Fisher Scientific, Waltham, USA). Proteins were separated by constant current (200 V) for 45–60 min using a Bio-Rad power supply. Immunoblotting was carried out with a Novex Semi-Dry Blotter (Invitrogen, Karlsruhe, Germany) and proteins were transferred to an activated (100% ethanol, 1 min; followed by two washing steps with

**Table 2.** Antibody information.
IHC, immunohistochemistry; IB, immunoblot.

| Antigen | Host species | Method, dilution | Source and Cat# |
|---|---|---|---|
| α-Actin | Mouse | IB 1:1000 | Millipore |
| α-APP | Mouse | IHC 1:1000 | Chemicon (#MAB348) |
| α-CAII | Rabbit | IB 1:500, IHC 1:300 | *Ghandour et al., 1980* |
| α-CMTM5 | Rabbit | IB 1:1000 | ProteinTech (custom made) Sequence: YRTELMPSTTEGD |
| α-CMTM5 | Rabbit | IHC 1:200 | Pineda (custom made) Sequence: CAFKIYRTELMPSTTEGDQQ |
| α-CNP | Mouse | IB 1:1000 | Sigma-Aldrich (#SAB1405637) |
| α-GFAP | Mouse | IHC 1:200 | Novo Castra (#NCL-L-GFAP-GA5) |
| α-IBA1 | Goat | IHC 1:1000 | Abcam (#ab5076) |
| α-MAC3 | Rat | IHC 1:400 | Pharmingen (#553322) |
| α-PLP | Rabbit | IB 1:2000 | A431 *Jung et al., 1996* |
| α-RBPMS | Guinea pig | IHC 1:300 | Sigma-Aldrich (#ABN1376) |
| α-SIRT2 | Rabbit | IB 1:500 | Abcam (#ab67299) |
| α-TUJ1 | Mouse | IHC 1:1000 | Covance (#MMS-435P) |
| α-mouse HRP | Goat | IB 1:10,000 | Dianova (#115-03-003) |
| α-rabbit HRP | Goat | IB 1:10,000 | Dianova (#111-035-003) |
| α-rabbit Alexa555 | Donkey | IHC 1:1000 | Dianova (#SBA-3030-32) |
| α-guinea pig Alexa555 | Donkey | IHC 1:1000 | Dianova |
| α-mouse STAR-RED | Goat | IHC 1:200 | abberior (# STRED-1001-500UG) |
| α-rabbit STAR ORANGE | Goat | IHC 1:200 | abberior (# STORANGE-1002-500UG) |

water) PVDF membrane (GE Healthcare, Buckinghamshire, UK; Cat# 10600023) at 20 V for 45 min. After blotting, membranes were blocked in 1× TBS containing 5% non-fat dry milk (Frema, Karlsruhe, Germany) and 0.05% Tween-20 for 1 hr at room temperature (RT). Primary antibodies were diluted in 5 ml blocking buffer and incubated overnight at 4°C and horizontal rotation. Membranes were washed thrice with TBS-T for 5–10 min each and incubated for 1 hr with secondary HRP antibodies diluted in blocking buffer. Membranes were washed three times with TBS-T for 5–10 min. Detection was carried out using enhanced chemiluminescent detection (ECL) according to the manufacturer's instructions (Western Lightning Plus-ECL or SUperSignal West Femto Maximum Sensitive Substrate; Thermo Fisher Scientific, St Leon-Rot, Germany). Immunoblots were scanned using ECL Chemostar (Intas Science Imaging, Göttingen, Germany). For antibody information, see *Table 2*.

## Label-free quantification of myelin proteins

In-solution digestion of myelin proteins according to an automated filter-aided sample preparation (FASP) protocol (*Patzig et al., 2016*) and LC-MS analysis by different MS$^E$-type data-independent acquisition (DIA) mass spectrometry approaches was performed as recently established for PNS (*Siems et al., 2020*) and CNS (*Jahn et al., 2020*) myelin. Briefly, protein fractions corresponding to 10 μg myelin protein were dissolved in lysis buffer (1% ASB-14, 7 M urea, 2 M thiourea, 10 mM DTT, 0.1 M Tris pH 8.5) and processed according to a CHAPS-based FASP protocol in centrifugal filter units (30 kDa MWCO, Merck Millipore). After removal of the detergents, protein alkylation with iodoacetamide, and buffer exchange to digestion buffer (50 mM ammonium bicarbonate [ABC], 10% acetonitrile), proteins were digested overnight at 37°C with 400 ng trypsin. Tryptic peptides were recovered by centrifugation and extracted with 40 μl of 50 mM ABC and 40 μl of 1% trifluoroacetic

acid, respectively. Combined flow-throughs were directly subjected to LC-MS analysis. For quantification according to the TOP3 approach (*Silva et al., 2006*), aliquots were spiked with 10 fmol/µl of Hi3 EColi standard (Waters Corporation), containing a set of quantified synthetic peptides derived from *Escherichia coli* chaperone protein ClpB.

Nanoscale reversed-phase UPLC separation of tryptic peptides was performed with a nanoAcquity UPLC system equipped with a Symmetry C18 5 µm, 180 µm × 20 mm trap column and an HSS T3 C18 1.8 µm, 75 µm × 250 mm analytical column (Waters Corporation) maintained at 45°C. Peptides were separated over 120 min at a flow rate of 300 nl/min with a gradient comprising two linear steps of 3–35% mobile phase B (acetonitrile containing 0.1% formic acid) in 105 min and 35–60% mobile phase B in 15 min, respectively. Mass spectrometric analysis on a quadrupole time-of-flight mass spectrometer with ion mobility option (Synapt G2-S, Waters Corporation) was performed in the dynamic range-enhanced-UDMS$^E$ mode as established previously for proteome analysis of purified myelin (*Jahn et al., 2020*; *Siems et al., 2020*). Continuum LC-MS data were processed using Waters ProteinLynx Global Server (PLGS) and searched against a custom database compiled by adding the sequence information for *E. coli* chaperone protein ClpB and porcine trypsin to the UniProtKB/Swiss-Prot mouse proteome (release 2017-07, 16,909 entries) and by appending the reversed sequence of each entry to enable the determination of false discovery rate (FDR). Precursor and fragment ion mass tolerances were automatically determined by PLGS and were typically below 5 parts per million (ppm) for precursor ions and below 10 ppm (root mean square) for fragment ions. Carbamidomethylation of cysteine was specified as fixed and oxidation of methionine as variable modification. One missed trypsin cleavage was allowed. Minimal ion matching requirements were two fragments per peptide, five fragments per protein, and one peptide per protein. The FDR for protein identification was set to 1% threshold.

For post-identification analysis including TOP3 quantification of proteins, ISOQuant (*Distler et al., 2013*; *Kuharev et al., 2015*) software freely available at http://www.immunologie.uni-mainz.de/isoquant/ was used as described previously (*Jahn et al., 2020*; *Siems et al., 2020*). Only proteins represented by at least two peptides (minimum length six amino acids, score ≥5.5, identified in at least two runs) were quantified as ppm, that is, the relative amount (w/w) of each protein in respect to the sum over all detected proteins. FDR for both peptides and proteins was set to 1% threshold and at least one unique peptide was required. Proteome profiling comparing myelin from *Cmtm5* cKO and CTRL mice was performed with three biological replicates and duplicate digestion, resulting in a total of six LC-MS runs per condition. The mass spectrometry proteomics data have been deposited to the ProteomeXchange Consortium via the PRIDE (*Perez-Riverol et al., 2019*) partner repository with dataset identifier PXD029443.

## Electron microscopy

For TEM, optic nerves and spinal cords were dissected and fixed in Karlsson-Schulz fixative (4% PFA, 2.5% glutaraldehyde in 0.1 M phosphate buffer [PB]) overnight. Samples were processed and embedded in epoxy resin (Serva, Heidelberg, Germany) as described (*Möbius et al., 2010*). For TEM, ultrathin (50 nm) sections were prepared using a PTPC Powertome Ultramicrotome (RMC, Tucson, AZ, USA) and a diamond knife (Diatome AG, Biel, Switzerland). Sections were cut and collected on formwar-coated copper grids (AGAR Scientific, Essex, UK). To enhance contrast, ultrathin sections were stained with UranyLess (Electron Microscopy Science, Hatfield, Panama) for 20 min and washed six times with $_{dd}H_2O$. For analysis, 16–20 nonoverlapping random images were taken per animal using the Zeiss EM900 at 7000× (one image = 220 µm²). All image analyses were performed using Fiji (version 2.0.0-rc-68/1.52i; *Schindelin et al., 2012*).

To quantify the relative number of pathological myelin sheaths (*Figure 2E and J*), all profiles on the micrographs were assessed and classified as pathological myelin if displaying myelin outfoldings, double myelination, or inner-tongue swellings. To quantify pathological axons (*Figure 3B and E*), all profiles on the micrographs were assessed and classified as pathological if displaying axonal swellings, tubovesicular structures, and amorphous axoplasm in an axon, absence of an identifiable axon in a myelin sheath or myelin whorls. For axon diameter analysis, all normal-appearing, accurately cross-sectioned myelinated axons were evaluated on 16–20 nonoverlapping random images. Data is presented as mean axonal diameter per animal. g-ratios were calculated as the ratio between axonal diameter and the outer diameter of the corresponding myelin sheath. In total, 180–200 axons were

randomly selected for g-ratio analysis from 16 to 20 EM images per mouse using the Fiji Grid tool (circular grids, 3 µm$^2$ per point, random offset).

For the analysis of axon numbers in optic nerves, semithin sections (thickness 500 nm) were cut and stained with methylene blue/azur II (1:1) for 1 min followed by a washing step with $H_2O$. Images were acquired at 100× using a bright-field light microscope (Zeiss AxioImager Z1 coupled to a Zeiss Axio Cam MRc camera; controlled and stitched by Zeiss Zen 1.0 software). Using Fiji, optic nerve images were separated into 55 µm$^2$ rectangles. From all rectangles filled with ON tissue, five were chosen at random and all axons were counted. Axon number is shown as the mean of five assessed rectangles per mouse.

## Focused ionbeam-scanning electron microscopy

Samples were prepared according to *Steyer et al., 2020*. In brief, dissected optic nerve samples were immersed in primary fixative (Karlsson-Schultz PB: 109.5 mM $NaH_2PO_4 \cdot H_2O$, 93.75 mM $Na_2HPO_4 \cdot 2H_2O$, 86.2 mM NaCl, 2.5% glutaraldehyde, 4% formaldehyde; adjust the pH to 7.4 and filter, at 4°C for at least 24 hr) and processed with a modified osmium-thiocarbohydrazide-osmium (OTO) protocol as previously described (*Weil et al., 2018*) based on a previously established original protocol (*Deerinck et al., 2010*). Briefly, samples were post-fixed for 3 hr with 2% $OsO_4$ (EMS, Hatfield, USA) and 1.5% $K_3Fe(CN)_6$ (EMS) at 4°C followed by a contrasting step with 1% thiocarbohydrazide (Sigma-Aldrich, St. Louis, USA) for 1 hr at RT and 1.5 hr incubation with 2% $OsO_4$. En bloc staining was performed with 2% uranyl acetate overnight at 4°C. The next day the samples were dehydrated through a series of ascending concentrations of acetone (EMS) for 15 min each (30%, 50%, 75%, 90%, 3 × 100%) and incubated with increasing concentrations of the epoxy resin Durcupan (Sigma-Aldrich) (2:1, 1:1, 1:2) for 2 hr each and left overnight in 90% Durcupan without component D. The next day the samples were incubated with 100% Durcupan (all components: A [epoxy resin] 11.4 g, B [hardener] 10 g, C [accelerator] 0.3 g, D [plasticizer] 0.1 g) for 4.5 hr and polymerized for 48 hr at 60°C.

The polymerized samples were trimmed using a 90° trimming knife (Diatome AG) and positioned on a SEM-stub using silver conductive resin (EPO-TEK 129-4) (EMS). The surface was sputter coated (Leica, ACE 600) (Leica, Wetzlar, Germany) with a layer of 10 nm platinum and placed inside the FIB-SEM (Crossbeam 540, Zeiss, Oberkochen, Germany). After exposing a cross-section through the region of interest with 15 nA ion current and polishing with 7 nA, a 400 nm deposition of platinum was performed using 3 nA. The final dataset was acquired at 1.5 kV (1000 pA) 5 nm × 5 nm × 25 nm voxel size with a milling current of 1.5 nA. Fiji (*Schindelin et al., 2012*) was used for all the following image processing steps: the images were aligned using the SIFT algorithm, cropped, and inverted. They were smoothed using a Gaussian blur (sigma 1) and a local contrast enhancement was applied (CLAHE: block size 127, histogram bins 256, maximum slope 1.25). The dataset was binned by 2 in x and y. Analysis of pathological myelin and axon profiles was carried out using Fiji. All profiles in the volume belonging to one of the categories (myelin outfoldings, inner-tongue inclusions, axoplasmic inclusions, myelin whorls) were counted, and values were normalized to a volume of 10,000 µm$^3$. Example 3D models were reconstructed using IMOD (v 4.9.12, University of Colorado, https://bio3d.colorado.edu/imod).

## Immunohistochemistry

Sections (5 µm) of paraffin-embedded brains were used to determine neuropathology and oligodendrocyte number. Section preparation was as previously described (*de Monasterio-Schrader et al., 2012*; *Patzig et al., 2016*). To assess neuropathology in *Cmtm5* cKO and control mice, 3–5 mice per genotype were analyzed for each timepoint (P30, P75, P365) and labeled for amyloid precursor protein (APP) (Chemicon, 1:1000), MAC3 (Pharmingen, 1:400), glial fibrillary acidic protein (GFAP) (Novo Castra, 1:200), or IBA1 (Abcam, 1:1000). Images were acquired at ×40 magnification using a bright-field light microscope (Zeiss AxioImager Z1, coupled to Zeiss AxioCam MRc Camera; controlled and stitched by Zeiss Zen 1.0 software). The hippocampal fimbria was analyzed by counting APP-positive axonal swellings per selected area or by using an ImageJ plugin to semiautomatically determine the area of GFAP/MAC3/IBA1 immunopositivity as previously described (*de Monasterio-Schrader et al., 2012*; *Lüders et al., 2017*; *Patzig et al., 2016*). To assess oligodendrocyte number, sections of paraffin-embedded brains from *Cmtm5* cKO and control mice at P75 were deparaffinized and rehydrated as with sections for neuropathology. Sections were then blocked for 1 hr at RT with

PBS containing BSA and horse serum. Incubation with primary antibody was then carried out over 48 hr at 4°C with anti-CAII antibody (1:300 in PBS containing 1% HS). Slides were washed thrice for 10 min with PBS and incubated with DAPI (Thermo Scientific, Waltham, USA) and anti-rabbit Alexa555 (Dianova, 1:1000 in PBS containing 1% HS). Slides were washed again thrice with PBS for 10 min and mounted using AquaPolymount. The hippocampal fimbria was imaged using the Axio Observer Z2 (Zeiss) at a ×40 magnification and stitched using Zeiss Zen 2011. All cells positive for both DAPI and CAII were identified as oligodendrocytes. All positive cells were counted and normalized to an area of 1 mm². For antibody information, see *Table 2*.

## Preparation of cryosections and confocal imaging

Cryosections were obtained from spinal cords immersion fixed with 4% PFA overnight. Nerves were then transferred to a sucrose buffer (10% [w/v], 20% [w/v], 30% [w/v] in 0.1 M PB) overnight at 4°C for each concentration. The nerves were then embedded in small plastic chambers using Tissue-Tek O.C.T. Compound (Sakura, Staufen, Germany). Nerves were stored at –20°C until further use. 10-μm-thick cross-sections were prepared using a cryostat (Reichert Jung Cryocut 18000, Wetzlar, Germany) and transferred to Superfrost Plus microscope slides (Thermo Fisher Scientific). Slides were dried for 30 min at RT and stored at –20°C until further use.

Sections were stained with the following protocol using 200 μl volumes per slide: 3 min methanol, 30 min permeabilization using PBS with 0.4% [v/v] Triton-X100 (Sigma-Aldrich) followed by blocking using DAKO blocking buffer (DAKO, Hamburg, Germany) for 60 min. TUJ1 (Covance) and CMTM5 (custom made by Pineda, Berlin, Germany; *Table 2*) antibodies were diluted in antibody diluent (DAKO) and incubated for 48 hr at 4°C in a dark, humid chamber. Sections were then washed three times with PBS for 10 min and incubated with secondary antibodies (α-rabbit STAR-RED, α-mouse STAR-ORANGE, abberior, Göttingen, Germany; *Table 2*) diluted 1:200 in antibody diluent for 60 min. Sections were washed three times with PBS for 10 min and mounted using AquaPolymount (Poly-sciences, Warrington, USA). Images were obtained on a Confocal and STED FACILITY line microscope (Abberior Instruments, Germany) and acquired as xy-plane with a pixel size of 30 nm. The fluoro-phores were excited with appropriate excitation lasers at 640 nm (abberior STAR-RED) and 561 nm (abberior STAR ORANGE). For image acquisition, the microscope software 'Lightbox' as provided by Abberior Instruments was used.

## Magnetic resonance imaging and spectroscopy

MRI and MRS were acquired on a 9.4 T Bruker BioSpec MR system with a 30 cm horizontal bore and B-GA12 gradient system operating on Bruker ParaVision 6.0.1 (hardware and software from Bruker BioSpin MRI GmbH, Ettlingen, Germany). A four-channel (2 × 2) receive-only mouse head coil was used, in combination with a 112/84 resonator, to acquire MRI and MRS (both from Bruker BioSpin MRI GmbH).

The MRI protocol included magnetization transfer (MT)-weighted images and diffusion-weighted images (DWIs). For MT, a 3D fast low-angle shot (FLASH) sequence was used to acquire three data-sets: MT-weighted, proton density-weighted, and T1-weighted (repetition time [15.1, 15.1, 18] ms, echo time 3.4 ms, flip angles [5°, 5°, 25°], two averages, voxel size 100 μm × 100 μm × 100 μm, acqui-sition time 18.4 min). These datasets were used to estimate MT saturation (MTsat) according to the method described by *Helms et al., 2008*. DWIs were acquired using a spin-echo echo-planar imaging sequence (repetition time 2000 ms, echo time 21.5 ms, two repetitions, voxel size 100 μm × 100 μm × 500 μm, gradient duration and separation 2.5 ms and 12.5 ms, b values 0, 1000, and 2000, gradient directions 30 for each b value, acquisition time 17.2 min). These DWIs were preprocessed through denoising (*Fadnavis et al., 2020*) and averaged across repetitions. A diffusion tensor model (*Basser et al., 1994*) was fitted to the preprocessed DWI data, and fractional anisotropy (FA), axial diffusivity (AD), radial diffusivity (RD), and mean diffusivity (MD) maps were derived (*Garyfallidis et al., 2014*). Multi-echo gradient-recalled echo (GRE) images were acquired using a 3D GRE sequence (repetition time 25 ms, echo time 2.2 ms, echo spacing 2 ms, number of echoes 10, flip angle 12°, four averages, voxel size 70 μm × 70 μm × 300 μm, acquisition time 19 min). The effective transverse relaxation rate ($R_2^*$) maps were calculated by fitting the multi-echo GRE magnitude signal decay across all echo times with a mono-exponential model (*Pei et al., 2015*).

MRS was acquired from cortices and corpus callosi using a stimulated echo acquisition mode (STEAM) sequence. The parameters of the STEAM sequence were repetition time of 6000 ms, echo time of 10 ms, spectral width of 5000 Hz, 2048 data points, 128 averages, and a total acquisition time of 12:48 min. The dimensions of the cortical and corpus callosal voxels were $3.9 \times 0.7 \times 3.2$ mm$^3$ and $3.9 \times 0.7 \times 1.7$ mm$^3$, respectively. All spectra were acquired with CHESS water suppression and outer volume suppression. The spectra were analyzed and quantified using LCModel (*Provencher, 1993*) in the chemical shift range from 0.2 to 4.2 ppm. Values with Cramer–Rao lower bounds above 20% were excluded from further analyses. Statistics were performed in Excel using a two-tailed, unpaired Student's *t*-test assuming equal variance.

## ERG and VEP

ERGs were recorded as described (*tom Dieck et al., 2012*), and VEPs were recorded essentially as described (*Ridder and Nusinowitz, 2006*). Briefly, mice were dark adapted overnight and anesthetized with ketamine (125 µg/g), xylazine (2,5 µg/g), and buprenorphine i.p. (0.1 mg/kg). Eyes were kept moist using contact lens solution containing hyaluronic acid for ERG recordings and with Methocel (DuPont Pharma, Mississauga, Canada) for VEP recordings. For ERG recordings, a silver ball electrode placed in the outer angle of the left eye served as active electrode. Signals were averaged 10 times. For VEP recordings, the scalp was resected and a small hole was drilled on the right side 1 mm lateral and 1 mm rostral of lambda and a thin needle electrode was inserted superficially. Signals were averaged at least 50 times. The reference electrode was placed on the nose of the mouse and the common ground near the hind legs. Signals were amplified 1000 times (NeuroAmp) and sampled without analog filtering. 0.1 ms light flashes were generated using BioSig Software and TDT system III hardware (Tucker Davis Technologies, Davis, USA) and presented via a custom-designed Ganzfeld apparatus at a stimulus rate of 0.5 Hz. Illumination was calibrated using a luxmeter (Mavolux 5032c, Nürnberg, Germany) and an Integrated Photodiode Amplifier 10530 (Integrated Photomatrix Limited, Dorchester, UK). Analysis was performed using custom-written MATLAB (version 2019b) scripts (*Figure 5—source code 1*). For analysis of ERG and VEP thresholds, Student's *t*-test was applied. Data from ERG and VEP analysis (*Figure 5*) is represented in line graphs showing mean values of mice per genotype ± SEM. To determine the genotype-dependent effect on ERG and VEP amplitude and latencies across various light intensities, two-way ANOVA was applied.

## Retina preparation and assessment of RGC number

To assess RGC numbers, eyes of 12-month-old *Cmtm5*$^{-/-}$ mice and respective controls were dissected and fixed for 1 hr with 4% PFA/PB. Eyes were then rinsed in PB and retinae were dissected as follows. The eye was cut open along the ciliary body and cornea and lens were removed. The retinal pigment epithelium was carefully removed. Four cuts on opposing sites were made to flatten the retina. The retina was transferred into a 24-well plate with one retina per well containing PBS. Retinae were washed with PBS/2% Triton X-100 (500 µl/well) at RT and gentle agitation for 10 min. To permeate nuclear membranes, the wash solution was replaced by fresh PBS/2% Triton X-100 and retinae were frozen at –80°C for 10 min. Retinae were washed twice with PBS/0.5% Triton X-100 for 5 min at RT. To reduce unspecific AB binding, retinae were incubated with blocking buffer (PBS/5% BSA/5% donkey serum/2% Triton X-100) for 1 hr at RT with gentle agitation. To label RGCs, retinae were incubated with guinea pig anti-RBPMS (Sigma-Aldrich; 1:200 in blocking buffer, 350 µl per well) for 2 hr at RT. Retinae were then washed thrice with PBS/0.5% Triton X-100 for 10 min at RT. RGCs were labeled using donkey anti-guinea pig Alexa555 (1:1000 in blocking buffer) and incubated overnight at 4°C. Retinae were then washed thrice for 30 min with PBS and transferred to a Superfrost slide with a fine brush. Retinae were mounted using AquaPolymount with the RGC layer facing up. Slides were kept at 4°C and dark until imaging. Images were taken using the Axio Observer Z2 (Zeiss) and a ×40 magnification and stitched using Zeiss Zen 2011. For assessment of RGC number, the average of three different areas (area = mm$^2$ per rectangle) was analyzed for each part of the retina (inner/middle/outer). Retinae of three individual mice per genotype were analyzed.

## Statistics

All experiments were analyzed blinded to genotypes. Statistical assessment was performed using GraphPad Prism 8 (GraphPad Software Inc, San Diego, USA) unless noted otherwise. Two-sided

Student's *t*-test was used to compare two groups unless specified otherwise. Welch's correction was performed in case of unequal distribution. Levels of significance were set as *p<0.05, **p<0.01, and ***p<0.001. Exact p-values are given in the figure legends, except those for the MRI data in *Figure 2—figure supplement 1a* are listed below. For all experiments, statistical test used and correction are given in the figure legends. Data in *Figure 1—figure supplement 1*, *Figure 2D, E, I, J, N*, *Figure 3B, C and E*, *Figure 3—figure supplement 1*, *Figure 3—figure supplement 2*, *Figure 3—figure supplement 3*, *Figure 4A"–D"*, *Figure 6C and F*, and *Figure 7C and E* are given as bar graphs with mean ± SEM; data points represent individual mice. Data from MRI analysis (*Figure 2—figure supplement 1*) and ERG and VEP thresholds (*Figure 5B and F*) are presented as boxplots; datapoints represent individual mice. Data for frequency distribution of axonal diameters (*Figure 3—figure supplement 1A'–C'*) are presented as bar graphs showing binned axonal diameters pooled of all mice per condition. Proteome data (*Figure 2B and B'*) is presented as volcano plot and heatmap. Data correspond to three mice per genotype and two technical replicate per mouse. The Bioconductor R packages 'limma' and 'q-value' were used to detect significant changes in protein abundance by moderated t-statistics as described (*Ambrozkiewicz et al., 2018*; *Siems et al., 2020*). Further information is provided in *Figure 2—source data 1*. g-ratios (*Figure 2G and L*) are presented as scatter plots. Each data point represents an individual axon. In total, 200 axons were analyzed per mouse and five mice were analyzed per genotype. Data from ERG and VEP analysis (*Figure 5*) is represented in line graphs showing the mean values of mice per genotype ± SEM. To determine the genotype-dependent effect on ERG and VEP amplitude and latencies across light intensities, two-way ANOVA was applied.

Exact sample size and number of mice are given in the figures or in the figure legends, except for the significance levels for MRI data in *Figure 2—figure supplement 1*, which were as follows. CC, corpus callosum; Fim, fimbria; Thal, thalamus; Cort, cortex; AC, anterior commissure. Two-sided Student's *t*-test was applied. (**B**) CC: p=0.1739; Fim: p=0.2244; Thal: p=0.3229; Cort: p=0.6159; AC: p=0.3290. (**C**) CC: p=0.7263; Fim: p=0.2223; Thal: p=0.9943; Cort: p=0.5009; AC: p=0.2425. (**D**) CC: p=0.7103; Fim: p=0.2608; Thal: p=0.6903; Cort: p=0.3576; AC: p=0.2531. (**E**) CC: p=0.4374; Fim: p=0.2038; Thal: p=0.8343; Cort: p=0.5728; AC: p=0.2678. (**F**) CC: p=0.7065; Fim: p=0.8432; Thal: p=0.7319; Cort: p=0.8614; AC: p=0.9983.

## Acknowledgements

We thank S Bode, K Kötz, A Fahrenholz, K Fuhrmann, D Hesse, R Jung, U Kutzke, and S Thom for technical assistance, G Hoch for hardware development and technical support, U Fünfschilling for in vitro fertilization, J Groh for providing protocols, S Ghandour and K Kusch for antibodies, U Suter and M Coleman for mice, and J Edgar for discussions. We thank the Wellcome Trust Sanger Institute Mouse Genetics Project (Sanger MGP) and its funders for providing the mutant mouse line *Cmtm5*[tm1a(KOMP)Wtsi].

## Additional information

### Competing interests

Martin Meschkat: MM is affiliated with Abberior Instruments Gmbh; the author has no financial interests to declare. Klaus-Armin Nave: Reviewing editor, eLife. The other authors declare that no competing interests exist.

### Funding

| Funder | Grant reference number | Author |
| --- | --- | --- |
| Deutsche Forschungsgemeinschaft | WE 2720/2-2 | Hauke B Werner |
| Deutsche Forschungsgemeinschaft | WE 2720/4-1 | Hauke B Werner |
| Deutsche Forschungsgemeinschaft | WE 2720/5-1 | Hauke B Werner |

| Funder | Grant reference number | Author |
| --- | --- | --- |
| European Research Council | Advanced Grant MyeliNano | Klaus Armin Nave |

The funders had no role in study design, data collection and interpretation, or the decision to submit the work for publication.

## Author contributions

Tobias J Buscham, Investigation, Visualization, Writing – original draft, Writing – review and editing; Maria A Eichel-Vogel, Anna M Steyer, Rakshit Dardawal, Tor R Memhave, Sophie B Siems, Christina Müller, Martin Meschkat, Ting Sun, Torben Ruhwedel, Investigation, Writing – review and editing; Olaf Jahn, Investigation, Visualization, Writing – review and editing; Nicola Strenzke, Investigation, Resources, Writing – review and editing; Wiebke Möbius, Investigation, Resources, Supervision, Writing – review and editing; Eva-Maria Krämer-Albers, Supervision, Writing – review and editing; Susann Boretius, Resources, Supervision, Writing – review and editing; Klaus-Armin Nave, Resources, Writing – review and editing; Hauke B Werner, Conceptualization, Funding acquisition, Project administration, Supervision, Writing – original draft, Writing – review and editing

## Author ORCIDs

Olaf Jahn http://orcid.org/0000-0002-3397-8924
Nicola Strenzke http://orcid.org/0000-0003-1673-1046
Wiebke Möbius http://orcid.org/0000-0002-2902-7165
Susann Boretius http://orcid.org/0000-0003-2792-7423
Klaus-Armin Nave http://orcid.org/0000-0001-8724-9666
Hauke B Werner http://orcid.org/0000-0002-7710-5738

## Ethics

Ethics StatementAll animal experiments were performed in accordance with the German animal protection law (TierSchG). Ethical review of animal experiments was performed by the Niedersächsisches Landesamt für Verbraucherschutz und Lebensmittelsicherheit (LAVES) and approved with licenses 33.19-42502-04-15/1833 and 33.8-42502-04-19/3172.

## Decision letter and Author response

Decision letter https://doi.org/10.7554/eLife.75523.sa1
Author response https://doi.org/10.7554/eLife.75523.sa2

# Additional files

## Supplementary files

- Transparent reporting form
- Source data 1. Source data displaying original immunoblots.

## Data availability

The mass spectrometry proteomics data have been deposited to the ProteomeXchange Consortium via the PRIDE partner repository with dataset identifier PXD029443. All other data generated during this study are included in the manuscript and supporting files. A Source Data file has been provided for Figure 2. A supporting file gives the numerical data used to generate the figures.

The following dataset was generated:

| Author(s) | Year | Dataset title | Dataset URL | Database and Identifier |
| --- | --- | --- | --- | --- |
| Jahn O | 2021 | Progressive axonopathy when oligodendrocytes lack the myelin protein CMTM5 | https://www.ebi.ac.uk/pride/archive/projects/PXD029443 | PRIDE, PXD029443 |

The following previously published datasets were used:

| Author(s) | Year | Dataset title | Dataset URL | Database and Identifier |
| --- | --- | --- | --- | --- |
| Jäkel S | 2019 | Altered oligodendrocyte heterogeneity in Multiple sclerosis | https://www.ncbi.nlm.nih.gov/geo/query/acc.cgi?acc=GSE118257 | NCBI Gene Expression Omnibus, GSE118257 |
| Zhou Y | 2020 | Human and mouse single-nucleus transcriptomics reveal TREM2-dependent and -independent cellular responses in Alzheimer's disease | https://www.ncbi.nlm.nih.gov/geo/query/acc.cgi?acc=GSE140511 | NCBI Gene Expression Omnibus, GSE140511 |

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
