## [Editor Report]

This manuscript by Buscham and colleagues investigates the phenotype of mice lacking the myelin protein chemokine-like factor-like MARVEL-transmembrane domain containing protein 5 (CMTM5). Loss of CMTM5 has no detectable effect on myelin structure, but nevertheless leads to axonal degeneration. The data reveal previously undescribed functions for a myelin protein in preserving CNS axonal integrity. This article will be of particular interest to those in the field of myelin biology as well as to neuroscientists interested in mechanisms underlying axonal function and degeneration.

---

## [Decision Letter]

**Decision letter after peer review:**

Thank you for submitting your article "Progressive axonopathy when oligodendrocytes lack the myelin protein CMTM5" for consideration by *eLife*. Your article has been reviewed by 3 peer reviewers, and the evaluation has been overseen by a Reviewing Editor and Gary Westbrook as the Senior Editor. The following individual involved in review of your submission has agreed to reveal their identity: Teresa L Wood (Reviewer #3).

Essential revisions:

1) Can the authors expand their analysis of spinal cord pathology to include axon loss, as was performed in the optic nerve? This request is based on the presumption that other tissues were preserved when optic nerves were harvested and could be used for these experiments without needing to generate new animals.

2) Given the discrepancy between axonal loss and RGC cell body preservation, the authors should perform additional analysis of synapses in the LGN or of Caspase-3, at least at the P75 time-point.

3) The genetic backgrounds of all mouse lines used needs to be clarified. The authors should also clarify whether heterozygous or homozygous Wlds mice were used for these studies.

4) A contribution of CMTM5 in other cell types cannot be ruled out from the experimental approaches employed and from published expression datasets indicating CMTM5's presence in astrocytes. If possible, improved IHC analyses or examination of CMTM5 in whole brain lysates could address this concern. At minimum, the text should be updated to discuss this caveat.

5) Figure 3C should refer to axon density rather than axon number.

6) Can the authors provide representative images of stainings done in Figure 3, supp 3?

7) Can the authors comment on CMTM5 expression in Schwann cells? An analysis of any PNS phenotypes is outside the scope of the present manuscript, but given the authors' previous work on CMTM6 in Schwann cells, it would be interesting to know whether or not CMTM5 is also in this cell type.

8) The Methods should be updated to define how the authors classified axons as pathological.

9) The authors should clarify if P24 vs 6m samples can be directly compared in Figure 1C,D

10) The authors should be consistent with weeks vs. days nomenclature for ages when discussing experimental endpoints.*Reviewer #1:*

Collectively, the data in this paper convincingly demonstrate that loss of CMTM5 in oligodendrocytes causes axonopathy in the absence of overt changes to myelin structure. This axonal pathology is demonstrated both at the histological as well as functional level (with Cmtm5 mutants showing a reduced amplitude of visually evoked potentials, consistent with axonal loss and/or dysfunction). The authors proactively address the potential concern that haploinsufficiency of CNP might sensitive axons to loss of CMTM5 by using both full Cmtm5 knockout and inducible conditional knockouts in addition to conditional knockouts using the CNP-Cre driver. Intriguingly, the axonal degeneration appears to occur without loss of neuronal cell bodies (at least in the retina) and is reduced by crossing the Cmtm5 mutants to Wlds. Although these observations are somewhat preliminary, they suggest that loss of Cmtm5 induces a Wallerian degeneration program in axons, which may be distinct from the degeneration seen in other myelin protein mutants. The paper does not currently address the mechanism by which loss of CMTM5 disrupts oligodendrocyte support of axons. Further investigation of this and whether the axonal loss mechanism is really different between Cmtm5, Plp1 and Cnp1 mutants will be important to establish in future work.

– The manuscript includes only a relatively superficial demonstration that the Cmtm5 phenotype is ameliorated by crossing the mice to the Wlds line (the partial rescue is only demonstrated using EM analysis of n=5 mice at a single time-point and CNS region). I understand that these are very time-consuming experiments to run given the time-points, but ideally analysis in other CNS regions would be performed.

– The RBPMS counts for RGCs may be underpowered to detect any loss (especially given the variability seen even within control mice). The authors' conclusion that there is a Wallerian process of axonal degeneration with preservation of the cell bodies is currently not very well supported. It's possible that staining retinal flatmounts or other CNS regions with markers of apoptosis (e.g., cleaved caspase 3) may be more sensitive in detecting any loss of neurons in the mutants, given the controls should show a very low background.

– Figure 3C is labelled as axonal number, but as the total area of the optic nerve is not accounted for should probably rather be referred to as axon density. (The reduction seen in mutant mice at 1 year of age is substantial enough that this is unlikely to be a substantial concern unless nerve size is substantially increased in the mutants through gliosis.)

– Ideally the genetic background would be described all the mouse lines used (particularly important if they vary between the Cmtm5 KO and WLDS mice or between different substrains of C57Bl/6 containing the Crb1rd8 mutation).

– The major weakness of the manuscript as it currently stands is the lack of mechanistic insight into how loss of Cmtm5 disrupts oligodendrocyte support of axons, somewhat reducing its impact. The comparison to other myelin mutants that cause axonopathy (CNP and PLP) is interesting. Nevertheless, I think it is still unclear whether loss of CMTM5 leads to axonal loss through a fundamentally different mechanism to these previously described mutants or whether the comparatively mild phenotype of the CMTM5 mutants rather represents a more modest version of the same pathological process. The paper should nevertheless be of interest to the myelin field and serve as the basis for future, more mechanistic, studies.*Reviewer #2:*

Following up with their work studying CMTM6 deletion in PNS Schwann Cells (SCs), the manuscript by Buscham et al., describes the effects of knocking out a CMTM6 paralog, CMTM5, from oligodendrocytes in the CNS. The authors find that while many parameters of myelination (i.e. myelin thickness, percent axons myelinated, etc.) as well as axon diameters remain normal (diameters are enlarged in SC-specific CMTM6 mutants), over time axons begin to show increasing signs of pathology, degeneration, and loss. Finally, they show that one of these pathological features (axon pathology observed in electron micrographs) is partially rescued when complete CMTM5 KOs are combined with Wallerian degeneration slow (Wlds) mice, suggesting that Wallerian Degeneration (WD) mechanisms are mediating the observed axon pathology.

The work presented is of great interest to the field and the manuscript shines with some very well done and thorough comparative analysis between conditional CMTM5 knock outs (cKOs) and controls. It adds CMTM5, for the first time, to a very small group of myelin proteins with key roles in maintaining axon integrity. However, it also has some weaknesses that stand in the way of the authors being able to make a strong conclusion about the WD mechanisms that may mediate the pathology observed. Since the authors cross complete CMTM5 KOs to Wlds mice to show the rescue, the authors cannot rule out involvement of CMTM5 loss from other cells. These conclusions can be greatly strengthened by performing the Wlds cross with oligodendrocyte-specific CMTM5 cKOs, as well as analyzing rescue of other pathological effects aside from axonopathy on electron micrographs. The manuscript can be further strengthened by improving immunohistochemical analyses, explaining discrepancies in axon loss versus lack of neuron loss, and examining PNS effects as describe below.

The authors make the claim that CMTM5 is a myelin protein as it is enriched in myelin containing fractions of brain lysate as well in mature oligodendrocytes in the brain. However, their immunohistochemical analysis in Figure 1B falls a bit short of showing CMTM5 expression in myelin. Co-localization of CMTM5 with a known myelin protein marker would greatly strengthen this conclusion.

In Figure 7, the authors show that there is no loss of RGC cell bodies despite the axon losses seen (Figure 3), which is surprising since axon loss often leads to neuron loss. How do the authors explain this apparent discrepancy? Is there evidence of synapse loss in the LGN?

Mice expressing CNPcre are used to conditionally delete CMTM5 from oligodendrocytes, but CNPcre is also expressed in Schwann Cells (SCs) therefore the mice the authors are examining will also have CMTM5 deleted from these cells. Is CMTM5 expressed in SCs? If so, is there similar axon pathology/degeneration in peripheral nerves? This would help to strengthen the authors' conclusions.

1) While it was great to see that immunohistochemical analyses for secondary neuropathological signs had been performed, showing representative images of the stainings done in Figure 3, supp 3 would be more supportive. It would be even better if these analyses could be performed in the optic nerve where the authors saw the most evidence for CMTM5 cKO-induced axon pathology.

2) It is not clear whether the Wlds mice used in the study are heterozygous or homozygous. If they are hets, would using homozygous mice reveal a greater rescue?

3) In the Materials and methods, a more detailed description of how the authors classified axons as "pathological" would be helpful. What specific characteristics did the authors look for (eg. darkened cytoplasm, large vesicles, swollen mitochondria, etc.) in the electron micrographs?

4) A very impressive and thorough examination of CMTM5 cKOs in this manuscript including the biochemical analysis, proteome analysis, electron micrograph analysis, MRI and MRS analyses, immunohistochemical analysis, FIB-SEM analysis, and functional analysis. Bravo!*Reviewer #3:*

The authors investigate oligodendrocyte function for a member of the chemokine-like factor-like MARVEL-transmembrane containing (CMTM) protein family, CMTM5. This builds on their prior findings showing that another member of the family, CMTM6, is necessary in Schwann cells, the myelinating cells of the peripheral nervous system (PNS), for normal function of peripheral axons. They hypothesized that CMTM5 might have a similar function in oligodendrocytes, the myelin-producing cells of the CNS, based on prior studies showing it is highly expressed in those cells and present in CNS myelin. Immunoblotting and microscopy validate prominent expression of CMTM5 in myelin. The authors also show convincing data for loss of the myelin CMTM5 in mice carrying a conditional deletion of Cmtm5 gene (cKO) specifically in oligodendrocytes. There are several interesting and important findings from this study. Of particular interest is the finding that CMTM5 is necessary for normal axonal integrity over time in the absence of any overt function in the formation of myelin developmentally which has been carefully analyzed in optic nerve. This is in contrast to the prior published results obtained from deletion of two other myelin proteins, CNPase and proteolipid protein (PLP). The authors nicely present these comparisons in Table format. The authors also show no loss of retinal ganglion neurons supporting the hypothesis that a Wallerian-type degeneration underlies the axonopathy. This is convincingly tested and demonstrated by introducing a WldS mutation (the Wld gene is necessary for Wallerian axon degeneration) into the Cmtm5 cKO mice which resulted in about a 50% reduction in the appearance of pathological axons. Moreover, loss of the Wld gene function reduces pathology in the mice with oligodendrocyte loss of Cmtm5 but not in mice where either Cnp or Plp are deleted from oligodendrocytes. Finally, the axon degeneration has modest functional consequences for the optic nerve: retinal function appears normal based on electroretinography recordings and visually evoked potentials indicating transmission of impulses to the visual cortex are intact but reduced in amplitude. The study overall reveals a new myelin protein involved in axon integrity in the absence of regulating developmental myelination or axon size.

The data presented are convincing and generally support the conclusions. There are a few questions raised by the study.

1) Although the data showing function in oligodendrocytes is convincing, the authors do not mention that the published expression data also show some expression in astrocytes (See Supp. Figure 1). Moreover, the Cmtm5 BAC-Cre described in Gong et al., 2007 was not rigorously tested for oligodendrocyte specificity. For the knockout validation in Figure 2, loss of CMTM5 is shown only in myelin but not in whole brain lysates as for Figure 1. It might be interesting to see these data. Is it possible astrocyte expression of CMTM5 serves a similar function for axons? It is outside the scope of this study but would be interesting to include in the discussion.

2) In Figure 2 there is a trend to an increase in percentage of unmyelinated axons at p30 in optic nerve but not significant. Is it possible this is due to decreased CNP? Because the deletion for most of the presented studies makes use of a mouse line with a knock-in of Cre recombinase into the Cnp locus, these mice also have 50% reduction in CNP protein. This might have potentially confounding effects on interpreting the phenotype. Although the Cnp heterozygous (Cnp+/-) mice were previously reported to lack the pathology seen in the Cnp-/- line, it is important for interpreting the myelin and axonal pathology data presented on the Cmtm5 cKO mice to clearly state if the same parameters, tissues and timepoints were analyzed in control Cnp+/- mice that are wild type for Cmtm5. It is noted, however, that the total KO of Cmtm5 shows axonal pathology independent of the reduction in Cnp expression. Also important for the conclusions is the demonstration that adult deletion of Cmtm5 using a tamoxifen inducible PLP-Cre results in a similar axonopathy phenotype.

3) The majority of data presented show optic nerve axon degeneration. In Figure 3 the authors show axonal pathology is increased and axon number is decreased at p365 in optic nerve. Here, dorsal spinal cord is also shown to have increased pathology but there are no data on whether axon number are decreased also in spinal cord. It would be interesting to know whether the pathology leads to loss of axons in other white matter tracks.

4) lines 131-132 should be modified from "…expression of CMTM5 in the CNS is largely limited to oligodendrocytes and enriched in CNS myelin" to eliminate "largely limited". The RNA expression data indicate although it is prominently expressed particularly in mature oligodendrocytes, it is also expressed in astrocytes. The protein expression in astrocytes was untested in this study but at least by RNA expression it is clearly expressed.

5) Figure 1C,D – were the samples P15-P24 run on separate blots from the samples 6m-24m? Is it possible to compare levels directly from P24 with 6m? It seems PLP is much reduced in the myelin fractions from the older animals but perhaps this comparison cannot be made. This should be clarified.

6) Figure 2 legend – correct spelling of individual in last sentence.

7) Figure 5 data showing differences in amplitude in the VEPs were done at 34 weeks (238 days) – no analysis of axonopathy was done at this time. It is a bit confusing for the reader to suddenly shift to 34 weeks for analysis – please include the day equivalent and probably modify the conclusion in the results that this is likely due to axonopathy occurring prior to the 365-day timepoint previously analyzed. The age should also be made clear in the figure legend.

---

## [Author Response]

Essential revisions:1) Can the authors expand their analysis of spinal cord pathology to include axon loss, as was performed in the optic nerve? This request is based on the presumption that other tissues were preserved when optic nerves were harvested and could be used for these experiments without needing to generate new animals.

In response, we have now quantified axonal density in spinal cord dorsal white matter at 12 months of age; the result is given in the new Figure 3F. We found a trend toward reduced axonal density in cKO, which however did not reach the threshold for significance (i.e. p≥0.05). We presume that reduced axonal density would gain significance with further progression upon further aging; yet to test this would require generating new animals and aging them for over 12 months.

2) Given the discrepancy between axonal loss and RGC cell body preservation, the authors should perform additional analysis of synapses in the LGN or of Caspase-3, at least at the P75 time-point.

In response, we performed the suggested Caspase-3 immunohistochemistry at P30, P75 and 12 months of age on coronal brain sections at the level of the hippocampal fimbria (n=5-6 mice per genotype, as for the other neuropathological markers shown in Figure 3-supplement 3). Both cortical grey matter and hippocampal fimbria (white matter) displayed a very low number of Caspase-3 immunopositive cells, ranging between zero and two per section and animal. We interpret this as absence of genotype-dependent increase in the number of Caspase-3-positive cells in mutants. We have abstained from including these negative data at this time. However, if specifically requested by the reviewers or Editors we will include the data, although they essentially show near absence of Caspase-3 positivity in all mice of both genotypes, at least at the assessed timepoints. However, we note that we can not exclude that a genotype-dependent difference in the number of Caspase-3 positive cells may emerge upon further aging; yet to test this would require generating new animals and aging them for over 12 months.

3) The genetic backgrounds of all mouse lines used needs to be clarified. The authors should also clarify whether heterozygous or homozygous Wlds mice were used for these studies.

In response we have now included a new paragraph into the ‘Mouse models and mouse lines‘ part of the ‘Material and methods‘ section, which gives the information about the genetic background. All *Wlds* mice were heterozygous; we have now further emphasized this information in the Results section and in the ‘Mouse models and mouse lines‘ part of ‘Material and methods‘.

4) A contribution of CMTM5 in other cell types cannot be ruled out from the experimental approaches employed and from published expression datasets indicating CMTM5's presence in astrocytes. If possible, improved IHC analyses or examination of CMTM5 in whole brain lysates could address this concern. At minimum, the text should be updated to discuss this caveat.

In response, we have performed the suggested immunoblot on whole brain lysates of control and cKO mice. The blot is included as the new Figure 2O. In this immunoblot, CMTM5 was readily detected in controls while it was virtually undetectable in cKO brains. Yet, given that astrocytes display some *Cmtm5*-expression at the mRNA level (as shown in Figure 1-supplement 1), we can not formally exclude that astrocytes may indeed also express some CMTM5 at the protein level (though not enough to yield a band in the cKO samples in the immunoblot). If so, it may certainly be functionally relevant. With regard to the detection of Cmtm5 mRNA in astrocytes, we have included in the Results section the statement:

“astrocytes also display a low level of *CMTM5*/*Cmtm5* mRNA (Figure 1-Supplement 1B, C, E).”

5) Figure 3C should refer to axon density rather than axon number.

We thank the reviewer for careful reading and have changed the term accordingly.

6) Can the authors provide representative images of stainings done in Figure 3, supp 3?

In response we have now included representative example images for the 12 months timepoint in Figure 3-supplement 3. If specifically requested by the reviewers or Editors we will also include examples for the P30 and P75 timepoints.

7) Can the authors comment on CMTM5 expression in Schwann cells? An analysis of any PNS phenotypes is outside the scope of the present manuscript, but given the authors' previous work on CMTM6 in Schwann cells, it would be interesting to know whether or not CMTM5 is also in this cell type.

Schwann cells indeed express CMTM5. In response, we have now added a short new paragraph to the Discussion section, in which we also cite three relevant publications (Patzig et al., 2011; Siems et al., 2020; Gerber et al., 2021).

8) The Methods should be updated to define how the authors classified axons as pathological.

In response, we thank the reviewer for careful reading and observing the lack of this information. We have now included the information in the Methods section as follows:

“To quantify pathological axons (Figure 3B, E), all profiles on the micrographs were assessed and classified as pathological if displaying axonal swellings, tubovesicular structures and amorphous axoplasm in an axon, absence of an identifiable axon in a myelin sheath or myelin whorls.”

9) The authors should clarify if P24 vs 6m samples can be directly compared in Figure 1C,D

In response, we have further widened the gap between the P15-P24 blots (left) and the 6m-24m blots (right), to even better convey that these come from different blots / membranes and can therefore not be directly compared. In addition, we presume this to be evident from the original blots being provided as separate source data files. We have also carefully re-checked our phrasing if it may imply a direct comparison but did not find such statement and thus did not change the text in this respect.

10) The authors should be consistent with weeks vs. days nomenclature for ages when discussing experimental endpoints.

In response, we now always give the age in days up to postnatal day 75 (P75), and in months for all higher ages.